# Sequential emergence and contraction of epithelial subtypes in the prenatal human choroid plexus revealed by a stem cell model

Haley Masters[1,2], Shuxiong Wang[3], Christina Tu[4], Quy Nguyen[5], Yutong Sha [2,3], Matthew K. Karikomi [3], Pamela Shi Ru Fung[1], Benjamin Tran[1], Cristina Martel[1], Nellie Kwang [1], Michael Neel[1], Olga G. Jaime[1], Victoria Espericueta[1], Brett A. Johnson [1], Kai Kessenbrock [5], Qing Nie [2,3] & Edwin S. Monuki [1,2,4] ✉

Despite the major roles of choroid plexus epithelial cells (CPECs) in brain homeostasis and repair, their developmental lineage and diversity remain undefined. In simplified differentiations from human pluripotent stem cells, derived CPECs (dCPECs) display canonical properties and dynamic motile multiciliated phenotypes that interact with Aβ uptake. Single dCPEC transcriptomes over time correlate well with human organoid and fetal CPECs, while pseudotemporal and cell cycle analyses highlight the direct CPEC origin from neuroepithelial cells. In addition, time series analyses define metabolic (type 1) and ciliogenic dCPECs (type 2) at early timepoints, followed by type 1 diversification into anabolic-secretory (type 1a) and catabolic-absorptive subtypes (type 1b) as type 2 cells contract. These temporal patterns are then confirmed in independent derivations and mapped to prenatal stages using human tissues. In addition to defining the prenatal lineage of human CPECs, these findings suggest dynamic models of ChP support for the developing human brain.

Inside the brain's ventricles, the choroid plexus (ChP) lies at the interface between two circulating fluids[1] – the blood and cerebrospinal fluid (CSF). At this interface, the ChP receives a disproportionate share of cerebral blood flow[2,3] via a rich network of fenestrated capillaries[1] and mediates a robust two-way exchange of blood and CSF components. In turn, the ChP produces the CSF (400–600 mL per day in humans)[1], which equilibrates with interstitial fluid of the brain and spinal cord across permeable linings (pia and ependyma) and via glymphatics[4]. CSF then returns to the peripheral circulation via arachnoid villi, granulations, and lymphatics in olfactory, dural, and basal regions[5]. These circulations and equilibria give the ChP and CSF access to CNS cells behind the blood-brain barrier, providing the rationale for CSF-based delivery of neurotherapeutics that cannot cross this barrier[6,7].

ChP functions at the blood-CSF interface are largely executed by its epithelial cells (CPECs)[1]. Among their roles, CPECs produce the CSF, secrete most of the CSF proteome, gate and transport molecules and immune cells between the blood and CSF, detoxify the circulating fluids, and form the anatomical blood-CSF barrier. Despite this broad physiologic and therapeutic relevance, remarkably little is known about human CPECs. The myriad of secretory and absorptive functions attributed to CPECs are thought to be carried out by a single epithelial cell type[1] that advances through developmental stages and asynchronous phases ("dark" and "light" cells[8,9]). Among CPECs, lineage

[1]Department of Pathology & Laboratory Medicine, University of California Irvine, Irvine, CA, USA. [2]Department of Developmental & Cell Biology, University of California Irvine, Irvine, CA, USA. [3]Department of Mathematics, University of California Irvine, Irvine, CA, USA. [4]Sue and Bill Gross Stem Cell Research Center, University of California Irvine, Irvine, CA, USA. [5]Department of Biological Chemistry, University of California Irvine, Irvine, CA, USA. ✉e-mail: emonuki@uci.edu

differences relative to the embryonic roof plate[10–12], and molecular differences based on ventricular origin have been described[13], and recent studies from mice[14] and from human ChP organoids[15] suggest a mammalian CPEC subtype specialized for ciliogenesis. However, formal delineation of CPEC subtypes and their lineage relationships remain lacking.

To address these basic and applied gaps, we previously developed methods[16,17] to generate derived CPECs (dCPECs) from mouse embryonic stem cells (ESCs) using an aggregate-organoid approach. (Note: "dCPEC" is used when referring specifically to the derived cells in vitro). Together with studies from mouse explants[18], the earlier derivations established fundamental developmental principles in mouse CPEC development, including BMP4 sufficiency as a CPEC morphogen[16,17] and the temporal restriction of CPEC competency to pre-neurogenic neuroepithelial cells (NECs) rather than radial glia[16]. For *human* dCPECs, we used human ESCs and the neural rosette method[19] to establish initial proof-of-concept[16], but this method had limited utility due to its high complexity, inefficiency, and inconsistency.

In this study, we describe and validate a human dCPEC protocol with enhanced simplicity, efficiency, consistency, and scalability. Using this protocol, we then define multiple forms of CPEC diversity in humans. We first describe reciprocal multiciliated and Aβ uptake phenotypes, as well as cilia motility, which change over time. We then use single cell RNA sequencing (scRNA-seq) across timepoints in vitro to define a human CPEC lineage tree with two bifurcations. Using independent derivations and perinatal tissues, we confirm and map the two bifurcations to midgestation and preterm periods in humans, then discuss the implications of this dynamic lineage to human brain development via two models.

## Results
### Simple, efficient, and accelerated human dCPEC generation from H1 ESCs
Our proof-of-concept method using human ESCs (H1 cells) yielded relatively few dCPECs (5–10%)[16]. To improve efficiency and simplicity, we optimized a feeder-free monolayer system involving the seeding of small ESC clumps on Matrigel followed only by media changes (Fig. 1A, B; Supplementary Fig. 1A, B). Compared to an alternative high-density method, which also worked efficiently (Supplementary Fig. 1C), the low-density seeding protocol provided ~30X greater scalability and up to 30 million dCPECs from 2 million starting ESCs.

To accelerate dCPEC generation, "rapid" neural induction media (NIM; Thermo Fisher) was introduced after ESC colonies reached 150–200 um in diameter (typically 2–3 days). As advertised, NIM led to ~100% dual positivity by immunocytochemistry (ICC) for neural progenitor markers (NESTIN and SOX2) and negativity for the pluripotency marker OCT4 by 7 days (Supplementary Fig. 1D). Timing and duration of NIM and BMP4 (10 ng/mL) were then co-optimized. Consistent with the early and short window of NEC competency for mouse CPEC fate[16], BMP4 addition 1 day in vitro (1 div) after NIM initiation, but not before (0 div) or after (2 div), resulted in substantial dCPEC induction (Fig. 1C). With BMP4 addition at 1 div, NIM for 5 days was most effective (Fig. 1D), after which the media was switched to CPEC media with BMP4. Like the mouse cells[16], induction efficiency was similar over a range of BMP4 exposure times (10–30 days; Supplementary Fig. 1E); 15 day BMP4 exposures were used for subsequent studies. Using this protocol, H1 ESCs, via an NEC progenitor, gave rise to dCPEC sheets and islands that routinely covered 40% or more surface area by 5–6 weeks (Fig. 1E; Supplementary Fig. 1F) and could remain adherent and viable past 1 year.

### dCPEC validations of canonical CPEC properties and functions
Like mouse dCPECs[16], human dCPEC islands formed three dimensional (3D) folds[20] and ridges over time and expressed established markers of several CPEC compartments by ICC (secretory apparatus/TTR[21], apical membrane/AQP1[22], basolateral membrane/AE2[23], cilia/ARL13B[24], nucleus/OTX2[25], tight junctions/CLDN1 and CLDN5[26]) in a stereotypical order (Fig. 1F; Supplementary Fig. 1G). The dCPECs also had a uniform apicobasal polarity, a well-known property of derived[16] and in vivo CPECs[1] (Supplementary Fig. 1H), with apical AQP1+ membranes facing the media while basolateral AE2+ membranes faced Matrigel or abutted one another within 3D folds (Fig. 1G). Comparable derivation efficiencies were achieved with multiple iPSC lines (Supplementary Fig. 1I–K). As in mouse cells[27], early cell cycle withdrawal was evident in human dCPECs, with relatively low EdU labeling after 21 div (Supplementary Fig. 1L) and almost none after 31 div (Fig. 1H).

Consistent with the high mitochondrial content of CPECs in vivo[28], dCPECs displayed high mitochondrial mass (ATPB; Supplementary Fig. 1M) and activity (MitoTracker Orange; Fig. 1I) compared to neighboring cells. CPECs are also well known to produce and secrete TTR via the classical secretory pathway[29]. By 1–2 h after washouts, TTR was readily detectable in culture media by ELISA (Fig. 1J) and blocked over a wide concentration range (0.1–10 ug/mL) of Brefeldin A (BFA), which inhibits the classical secretory pathway[30]. BFA blockade was also fully reversible (Fig. 1J), ruling out toxicity as an explanation for the TTR reductions.

### Dynamic multiciliated CPEC phenotypes in vitro and in vivo
CPECs are relatively unique among CNS cells in being multiciliated[31]. Even at low magnification, the cilia marker ARL13B readily distinguished monociliated non-dCPECs and neural rosettes[32] (Supplementary Fig. 2A) from the islands of dCPECs with multiple apical cilia per cell (Fig. 2A, B). Qualitative impressions of cilia mass increases per dCPEC between 38 and 80 div (Fig. 2B; Supplementary Fig. 2B, C) were confirmed using a metric based on ARL13B and ZO1 staining (Fig. 2C; Supplementary Fig. 2D; see Methods). In addition, cilia occupied increasing percentages of dCPEC surface area over this time period (Fig. 2B, D).

To assess cilia distributions in vivo, we evaluated whole mount preparations of human postmortem ChP using the same two antibodies (ARL13B and ZO1). Around midgestation (23 postconceptional weeks/pcw), multiciliated CPECs were abundant, and their cilia were generally clustered. Near term (38 pcw) and into early adulthood (19 years of age), CPECs remained multiciliated, but their cilia were widely dispersed (Fig. 2E; Supplementary Fig. 2E). Thus, cilia clustering and dispersion in vivo paralleled that seen in dCPECs over time in vitro.

### Interacting multiciliated and Aβ uptake phenotypes
Robust uptake (absorption) is another fundamental property of CPECs[1]. Uptake of Aβ peptides associated with Alzheimer's disease has been demonstrated for rat ChP[33], but not yet for human CPECs. After treatment with fluorescently-tagged Aβ$_{1-42}$ (AnaSpec), fluorescence in 64-div dCPECs was evident within 30 min, peaked by 6 h, then decreased markedly by 24 h (Fig. 2F, G; Supplementary Fig. 2F) across a range of concentrations (500 nM-4 uM). Aβ$_{1-42}$ fluorescence was intracytoplasmic (Supplementary Fig. 2G) within EEA1+ endosomes (Supplementary Fig. 2H), indicating rapid and robust Aβ$_{1-42}$ uptake into the early endosomes of human dCPECs. EEA1 colocalization also decreased over time (Supplementary Fig. 2H, I), suggesting Aβ$_{1-42}$ transfer from early endosomes to other intracellular compartments.

Across concentrations and timepoints, intracellular Aβ$_{1-42}$ appeared as variably-sized puncta and often in circular arrangements (Fig. 2F). ARL13B costaining revealed cilia at the center of these arrangements with Aβ-positive dCPECs having cilia that were generally more clustered (Fig. 2H). 1 uM or 2 uM Aβ$_{1-42}$ application itself resulted in dCPECs having more clustered cilia than vehicle controls, with peak clustering 1 h after application (Fig. 2I) and less by 2 h (Supplementary Fig. 2J). Moreover, Aβ$_{1-42}$ signal intensities were inversely correlated with cilia dispersion

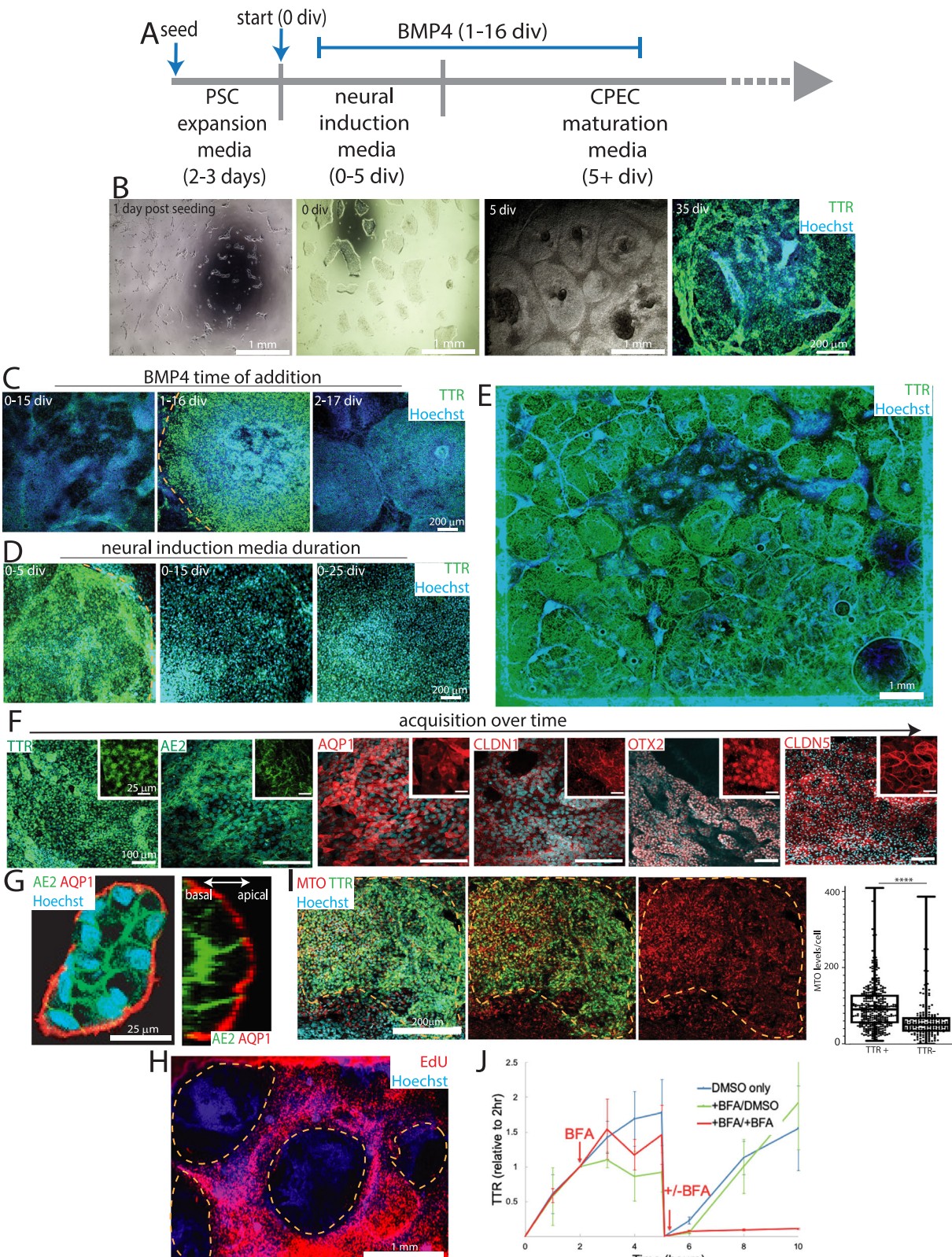

(Fig. 2J). These findings suggest reciprocal interactions between cilia clustering and Aβ$_{1-42}$ uptake in human dCPECs.

**Direct dCPEC origin from tripotent neuroepithelial cells: Pseudotemporal scRNA-seq analysis**

We then acquired four scRNA-seq datasets from 31 div (after apparent dCPEC cell cycle exit; Fig. 1H) to 75 div when dCPEC changes in

morphology and immunocytochemical profile appeared to decelerate (Fig. 1F). Timepoints were paired (31–46 and 55–75 div pairs; Fig. 3A) for culturing and processing through a 10x Genomics-Illumina pipeline to reduce batch effects, which we were unable to discern across the four datasets (Supplementary Fig. 3A, B). Datasets were processed primarily through SoptSC, which performs unsupervised clustering, pseudo-temporal ordering, and lineage inference in parallel[34], as well as Seurat[35].

**Fig. 1 | dCPEC derivation from H1 ESCs.** Dashed lines demarcate dCPEC islands. **A** Protocol schematic. Small ESC clumps seeded at low density and subjected only to media changes. BMP4 treatment is 15 days starting at 1 div. **B** Derivation milestones (phase contrast and epifluorescent TTR ICC) include small low-density ESC colonies after seeding, colony growth to 150–200 um diameter at the start of derivation (0 div), confluent islands by 5 div, and 3D islands by 35 div. **C** BMP4 time-of-addition (epifluorescent TTR ICC; 20 div). 15 day BMP4 application starting at 1 div, but not 0 or 2 div, leads to strong dCPEC induction. **D** Neural induction media (NIM) duration (epifluorescent TTR ICC; 25 div). Five days of NIM leads to better dCPEC differentiation than 15 or 25 days. **E** Example of an efficient derivation (epifluorescent TTR ICC; 45 div). Stitched images of one well in an 8-well chamber slide. **F** Marker acquisition order (confocal maximal projections; ICC): 35 div for TTR, AE2, AQP1, CLDN1; 50 div for OTX2; 70 div for CLDN5. **G** Apicobasal polarity (confocal and orthogonal views; ICC 45 div). Apical AQP1+ surfaces face the media, while AE2+ basolateral surfaces abut. **H** EdU incorporation (40 uM EdU; epifluorescent; 35 div). EdU application from 31–35 div leads to robust labeling of proliferating non-dCPECs (red), but not dCPEC islands. **I** Mitochondrial activity (confocal maximal projections of 150 nm MTO with TTR-ZO1 ICC; 35 div, ZO1 not shown; box plots designate medians, means "+", 25th-75th quartiles, and min-max ranges) showing TTR+ dCPECs with significantly higher MTO levels (red) than non-dCPEC neighbors (t-test $p < 0.0001$****; dCPEC $n = 367$ cells, non-dCPEC $n = 200$ cells). **J** TTR secretion into media following washouts (1 ug/mL BFA; ELISA; means +/- s.d.; 45 div). TTR increases in control wells (blue line) were reduced or blocked acutely by BFA at 2 h (red and green lines), then increased in the absence of BFA after secondary washouts at 5 h, indicating reversibility. Source data provided as a Source Data file. See Statistics and Reproducibility section of Methods for additional information.

SoptSC analyses suggested six cell clusters at 31 div (Supplementary Fig. 3C), which increased to eight by 75 div (Fig. 3B). Differentially-expressed genes (DEGs) and cell-type markers[13–16] identified single clusters of dCPECs, neurons, and neuroepithelial cells (NECs) along with multiple clusters of neural progenitors – three at 31 and 46 div, four at 55 div, and five at 75 div (Fig. 3B, C). The same populations, with similar DEGs, were identified by Seurat (Supplementary Fig. 3D–F). All defined cell clusters were neural in origin; no non-neural cell clusters were identified in any dataset, underscoring the efficiency and consistency of neural induction and differentiation using the simplified protocol.

Consistent with NECs being stem progenitors, NEC fraction was highest at the earliest time point (31 div), while dCPEC and neuron fractions peaked at 46 div before declining (Fig. 3C). At 46 div, dCPECs comprised ~30% of cells. By 55 and 75 div, dCPEC fractions decreased as additional classes of proliferating neural progenitors appeared, expanded, and contracted in an orderly fashion (Fig. 3C).

Pseudotemporal analyses of individual timepoints suggested three classes of NEC progeny. At all four timepoints, dCPECs arose directly from NECs as a direct and distinct end-branch (pink TTR-expressing cluster in Fig. 3D–F), as did neurons. Neural progenitors constituted the third progeny class, which had more varied relationships to each other and to NECs (Fig. 3D,F) consistent with their significant transcriptomic similarities overall. Among the three classes of NEC progeny (Fig. 3G), dCPECs had a consistently direct lineage relationship to NECs based on pseudotemporal ordering.

### Orderly dCPEC transcriptome changes compared to human fetal and organoid CPECs

The four scRNA-seq datasets were then compared to published human ChP organoid (55 div)[15] and fetal CPECs (8 and 14 pcw)[36]. With batch correction by reciprocal principal component analysis (RPCA)[37], CPECs from all sources clustered together and away from neurons and NECs by UMAP (Supplementary Fig. 4A) and principal component analysis (PCA) (Supplementary Fig. 4B). Batch correction using SCTransform[38] increased the differences among CPEC clusters, which nonetheless continued to cluster and correlate well by UMAP and PCA (Fig. 3H, I). Pearson correlations between sources, using either batch correction method, supported dCPEC identity (Fig. 3J; Supplementary Fig. 4C, D). These analyses also revealed increasing dCPEC-to-fetal CPEC correlations with increasing time in vitro (Fig. 3H–J; Supplementary Fig. 4D), while the organoid CPECs displayed an intermediate differentiation profile (Fig. 3I, J) consistent with their intermediate derivation time (55 div)[15].

### dCPEC subtypes and bifurcations: Pseudotemporal and time-series scRNA-seq analyses

Cells comprising the dCPEC lineage (dCPECs and NECs) were then re-run through SoptSC (Supplementary Fig. 5A, B). At the two early timepoints (31 and 46 div), two dCPEC sub-clusters were evident (Fig. 4A–C). The sub-cluster with greater similarity to NECs was designated CPEC "type 1" (C1); the other sub-cluster was designated CPEC "type 2" (C2) (Fig. 4B; Supplementary Fig. 5C). C2 fraction was highest at 31 div (~40% of dCPECs compared to ~10% at 46 div) (Supplementary Fig. 5D), and C2 cells were present at the two later time-points (Fig. 4D; Supplementary Fig. 5F–H), but were too few in number to define as clusters computationally (see Methods).

Type 1 dCPECs were present at all four timepoints, but branched into two sub-clusters in the last 75-div timepoint (Fig. 4A–E). The sub-cluster with higher Pearson correlation (98%) to 55-div C1 cells was designated CPEC "type 1a" (C1a); the other sub-cluster (82%) was designated CPEC "type 1b" (C1b) (Supplementary Fig. 5I). These correlations suggested the emergence of a distinct C1b subtype from the C1-C1a lineage. Similar sub-clusters and pseudotemporal relationships were obtained using Seurat (Supplementary Fig. 5J–L).

For time-series analysis, dCPEC and NEC sub-clusters were analyzed using STITCH[39]. The roughly-triangular STITCH dot plots (Fig. 4F–H; Supplementary Fig. 6A) suggested one broad dCPEC and two broad NEC groups. The dCPEC group contained dCPECs from all four timepoints (Fig. 4F–H; Supplementary Fig. 6A). Additional STITCH analyses illustrated clear C1-C2 and C1a-C1b separations, supporting their designations as distinct subtypes (Supplementary Fig. 6B). STITCH lineage trees were then formally constructed. At all time-points, C2 cells were distinguished from the C1 lineage and NECs. The C1 lineage spanned all four timepoints, then bifurcated into C1a and C1b subtypes at the last timepoint (Fig. 4F–H, lower panels). Taken together, the SoptSC and STITCH analyses suggest two temporally-separated bifurcations within a branched dCPEC lineage tree.

We explored whether previously-described differences among CPECs could account for the dCPEC subtypes. CPECs in mice[11,12] and humans[40] have differing lineage relationships with the embryonic roof plate, but roof plate markers[11] were not selectively associated with a dCPEC branch or subtype (Supplementary Fig. 6C). Distinct anteroposterior (ventricular) CPEC identities have been described in mice[13,14], but all dCPEC sub-clusters were mixtures of cells with lateral ventricle (LV), fourth ventricle (4V), or unspecified identities (Supplementary Fig. 6E). The favoring of LV over 4V identity (Supplementary Fig. 6E) would be consistent with the "default" model of neural induction, which favors anterior specification[41], as well as the lack of Sonic hedgehog (SHH) agonism in our protocol, which is implicated in the specification of 4V ChP identity[42]. Regardless, neither roof plate lineage nor ventricular identity explained dCPEC branches or subtypes.

### Unique dCPEC 'G0' signature and direct origin from tripotent NECs: Cell cycle analyses

C1a-C1b lineage bifurcation in vitro occurred well after the apparent cell cycle withdrawal of dCPECs (Fig. 1H). Using Seurat[35], which uses S, G2/M, and G0/G1 phase gene lists from mice[43], dCPECs displayed

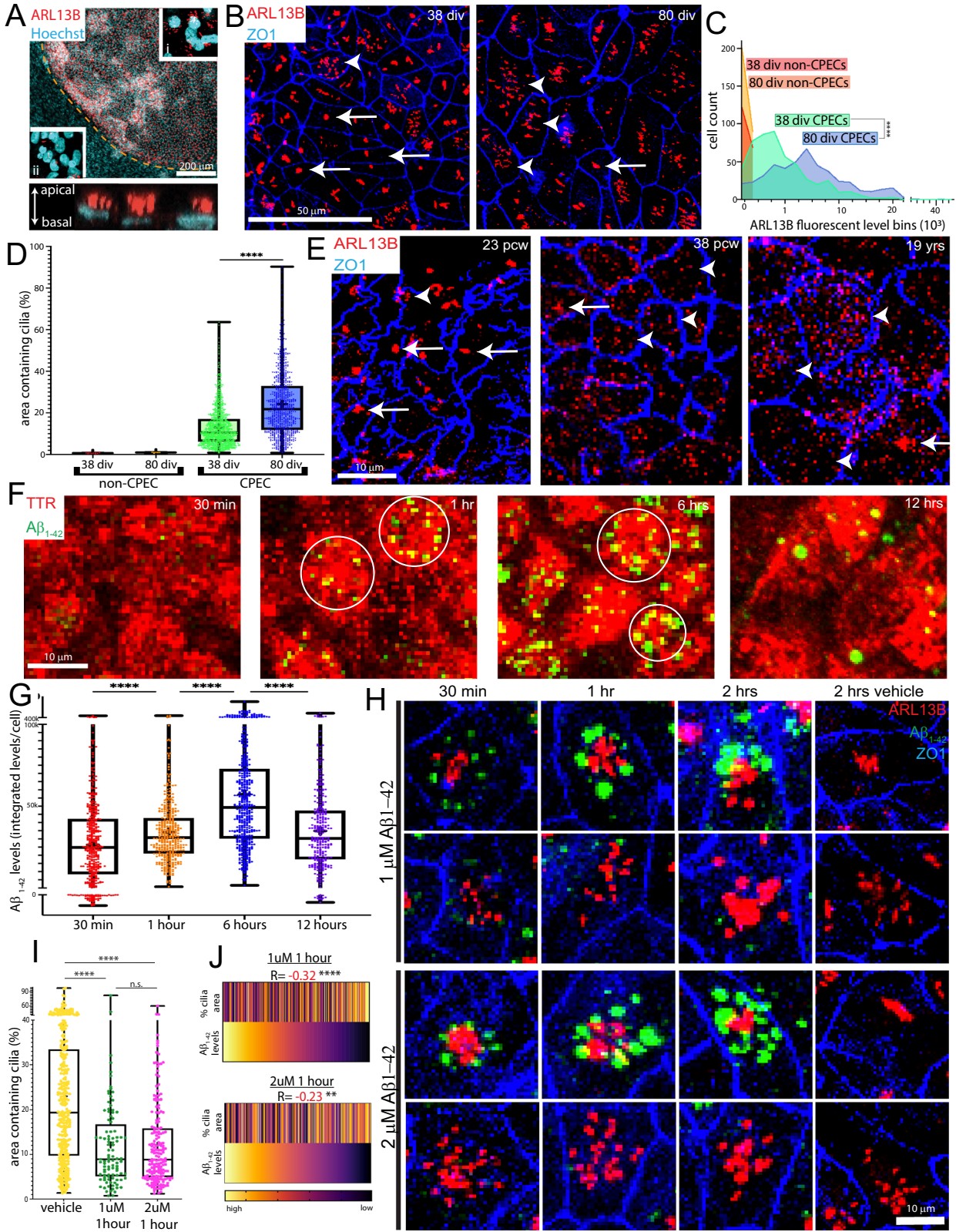

G0/G1 or G1/G2M signatures (Supplementary Fig. 7D); the G1/G2M dCPECs correspond to C2 cells due to the extensive overlap between ciliogenesis and G2/M genes[44] (see next section for more on ciliogenesis). Individual dCPECs from all timepoints formed a cluster distinct from NECs, neurons, and neural progenitors by UMAP (Supplementary Fig. 7D). When grouped by timepoint, dCPEC clusters separated from

the other cell types by PCA (Supplementary Fig. 7E). Interestingly, the three branches of NEC progeny (Fig. 3) were also clearly delineated by this cell cycle transcriptomic analysis alone (Supplementary Fig. 7E-F).

We then used human cell cycle genes that distinguish a 'neural' G0 phase from G1[45]. The dCPECs from all stages had predominant G0 signatures, as did neurons (Fig. 5A, B; Supplementary Fig. 7A, B),

**Fig. 2 | Dynamic and interacting multiciliated and Aβ$_{1-42}$ uptake phenotypes.**
Images are *en face* maximal projections except for the *in profile* image in A. Box
plots designate medians, means "+", 25th-75th quartiles, and min-max ranges.
**A** Multiciliated phenotype (ARL13B ICC; 35 div); dashed line demarcates dCPEC
island. Compared to monociliated non-CPECs (inset ii), the multiple apical cilia of
dCPECs (inset i and lower panel) are readily apparent even at low magnification.
**B–D** dCPEC cilia mass and dispersion over time (ARL13B-ZO1 ICC). dCPEC cilia are
more clustered at 38 div than 80 div (**B**). Cilia mass per dCPEC increases over time
compared to non-dCPECs (**C**) (one-way ANOVA $p = 0.0012$, Bonferroni corrected $t$-
test $p < 0.0001$****; 38-div $n = 530$, 80-div $n = 489$). dCPEC cilia-containing surface
area also increases over time (**D**) (one-way ANOVA $p < 0.0001$, Bonferroni cor-
rected $t$-test $p < 0.0001$****; 38-div $n = 1189$, 80-div $n = 923$). **E** CPEC cilia dispersion
in vivo (ARL13B-ZO1 whole mount ICC). Clustered cilia (arrows) are more frequent
than dispersed cilia (arrowheads) at 23 pcw, but uncommon at 38 pcw and 19 years
of age. **F, G** Aβ$_{1-42}$ uptake and time course (fluorescent Aβ$_{1-42}$ with TTR-ZO1 ICC; 64

div). Aβ$_{1-42}$ uptake (green) into TTR+ dCPECs (red) often occurs in circular
arrangements (circles in **F**), with uptake peaking by 6 h, then decreasing by 12 h (**G**)
(one-way ANOVA $p < 0.0001$, Bonferroni corrected $t$-tests $p < 0.0001$****; 30 min
$n = 419$ cells, 1 h $n = 415$ cells, 6 h $n = 492$ cells; 12 h $n = 300$ cells). **H–J** Aβ$_{1-42}$ uptake
and multicilia interaction (fluorescent Aβ$_{1-42}$ with ARL13B-ZO1 ICC; 64 div). 1 uM or
2 uM Aβ$_{1-42}$ uptake is robust by 1- or 2 h in dCPECs with clustered cilia (**H**). After 1 h
1uM or 2 uM Aβ$_{1-42}$ results in dCPEC populations with more clustered cilia com-
pared to vehicle controls (**I**) (one-way ANOVA $p < 0.0001$, Bonferroni corrected $t$-
tests ****$p < 0.0001$; $p = 0.97$ n.s.; vehicle $n = 352$, 1 uM $n = 100$, 2 uM $n = 184$). Rank-
order heatmaps (scale reflects a cells rank) anchored by Aβ$_{1-42}$ signal show negative
correlations between Aβ$_{1-42}$ levels and cilia clustering after 1 h in 1 uM or 2uM Aβ$_{1-42}$
(**J**) (Spearman correlations ****$p < 0.0001$, **$p < 0.01$). Source data provided as a
Source Data file. See Statistics and Reproducibility section of Methods for addi-
tional information.

---

but dCPECs and neurons were clearly distinguished from one another,
as well as from NECs and neural progenitors, by UMAP or PCA
(Fig. 5A–D). Some C2 cells again displayed a G2M signature that
nonetheless clustered with C1 cells (Fig. 5B,D). Notably, as with Seurat,
human neural G0 analysis alone delineated the three progeny classes
of tripotent NECs (Fig. 5C–E; Supplementary Fig. 7C). Neural G0 ana-
lysis of the earliest dataset alone (31 div) delineated these clas-
ses (Fig. 5F).

To confirm their non-cycling status, we stained derived and
perinatal CPECs for KI67, which labels cycling cells[46]. At a derivation
timepoint (49 div) before C1a-C1b lineage bifurcation (Fig. 4H), KI67
clearly stained non-CPECs, but neither dCPECs nor neurons (Fig. 5G;
Supplementary Fig. 7I). Similarly, in midgestation tissue (23 pcw)
before C1a-C1b lineage bifurcation (see next section), KI67-positive
cells could be seen in ChP stroma, but not in CPECs (Fig. 5H).

## Characterizations and confirmations of early type 1 and type 2 CPEC subtypes

All of the shared KEGG pathways[47] and GO terms[48] enriched in the top-
300 DEGs of C1 and C2 cells at 31 div (when C2 cells were most
abundant) were neurodegenerative diseases (Alzheimer, Huntington,
Parkinson, and prion diseases; Fig. 6A). Selective C1 enrichments were
associated with metabolism (e.g., metabolic pathways, non-alcoholic
fatty liver disease) and energy (e.g., oxidative phosphorylation, ther-
mogenesis), while C2 enrichments centered on ciliogenesis and cilia
maintenance (Fig. 6A, B). Inspection of the top-10 individual DEGs at 31
and 46 div further highlighted the strong ciliogenesis signature of C2
cells (Fig. 6C).

Multiciliated CPECs in zebrafish and mice are motile[49,50], and time-
lapse videomicroscopy revealed human dCPECs with highly motile
cilia (Supplementary Movies 1, 2). Tracking of individual 42-div dCPEC
cilia revealed heterogeneity in beat orientations, magnitudes (Fig. 6D),
and frequencies (0–10 Hz). In contrast, cilia motility was negligible in
otherwise healthy-appearing dCPECs by 110 div (Fig. 6E; Supplemen-
tary Movie 3). In transmission electron micrographs of 40-div dCPECs,
103 of 131 identified ciliary axonemes possessed the central pair of
singlet microtubules that distinguishes the 9 + 2 microtubule
arrangement of motile cilia[51] (Fig. 6F) along with other ultrastructural
features of multiciliation (Supplementary Fig. 8I).

Master regulators of motile ciliogenesis[52], particularly FOXN4[53]
and MCIDAS[54], were collectively higher in C2 compared to C1 cells at all
four timepoints (Fig. 6G, H; Supplementary Fig. 8F). Secondary analysis
of scRNA-seq data from human ChP organoids[15] also identified a CPEC
subpopulation that coexpresses FOXN4 and MCIDAS (Supplementary
Fig. 6F). FOXN4 and MCIDAS protein expression were then examined
in independent derivations. Both were expressed at high (Supple-
mentary Fig. 8A) and highly-correlated levels (Supplementary Fig. 8E)
in early dCPECs (31 div) with FOXN4 demonstrating particularly

discrete "salt and pepper" expression (Fig. 6I; Supplementary
Fig. 8A, B). Over time, FOXN4-expressing C2 cells remained detectable,
but significantly reduced in fractional positivity and expression level
(Fig. 6I; Supplementary Fig. 8C). Likewise, MCIDAS expression levels
decreased over derivation time (Supplementary Fig. 8A).

We then examined perinatal ChP tissues using the same two
antibodies (FOXN4 and MCIDAS). Around midgestation (23 pcw),
MCIDAS levels correlated positively with those of FOXN4 (Supple-
mentary Fig. 8F) with MCIDAS demonstrating particularly discrete and
mosaic expression in a substantial CPEC fraction (Fig. 6J). Near term
(42 pcw), MCIDAS-expressing C2 cells were detectable, but sig-
nificantly reduced in fractional positivity and expression level (Fig. 6J;
Supplementary Fig. 8D). Similarly, FOXN4 expression levels decreased
from midgestation to term (Supplementary Fig. 8G). Consistent with
the scRNA-seq and ICC findings in vitro, the perinatal tissues demon-
strate higher type 2 CPEC prevalence at midgestation, which decreases
significantly by term.

## Characterizations and confirmations of late type 1a and type 1b CPEC subtypes

Based on the top-300 C1a and C1b DEGs, C1b cells were uniquely
enriched for endocytotic, stress response, and catabolic KEGG path-
ways (Fig. 7A) and Gene Set Enrichment Analysis (GSEA)[55] terms
(Fig. 7B). Conversely, while also sharing several pathways with the
earlier-stage C1 cells (Fig. 7A–C), C1a cells were selectively enriched for
several secretory, water transport, and anabolic pathways based on
manually-curated GSEA gene lists and GSEA plots (Supplementary
Fig. 9A). GSEA analyses also revealed additional enrichments in C1b
cells, such as immune cell signaling interactions and steroid hormone
biosynthesis (Supplementary Fig. 9B; Supplementary Data 1). C1a and
C1b cells continued to share neurodegenerative disease pathways
(Fig. 7A). Pearson correlations between top C1a and C1b DEGs were
similar between 31 and 46 div, increased for a subset at 55 div, then
strongly diverged by 75 div (Fig. 7D, E). Thus, 55-div C1 cells had fea-
tures of a primed or mixed intermediate in normal development and
transdifferentiation[56]. Re-examinations of the human fetal[36] and orga-
noid datasets[15] did not reveal C1a and C1b subtypes (Supplementary
Fig. 6F) consistent with C1a-C1b emergence after the stages sampled in
these studies.

Specification of C1a and C1b subtypes was then examined in
paired derivations and human tissues using antibodies against CLDN5
(C1a) and SC5D (C1b) (Fig. 7E). While also known for its expression by
endothelial cells, CLDN5 has been described in human[57] and zebrafish
CPECs[58], although not in mouse CPECs[14], raising the possibility of
species-specific differences. At 61 div, correlation between CLDN5 and
SC5D trended positively, then became significantly negative by 79 div
(Fig. 7F–H; Supplementary Fig. 9D–H, L). By 79 div, C1a and C1b sub-
types also appeared clustered rather than randomly distributed

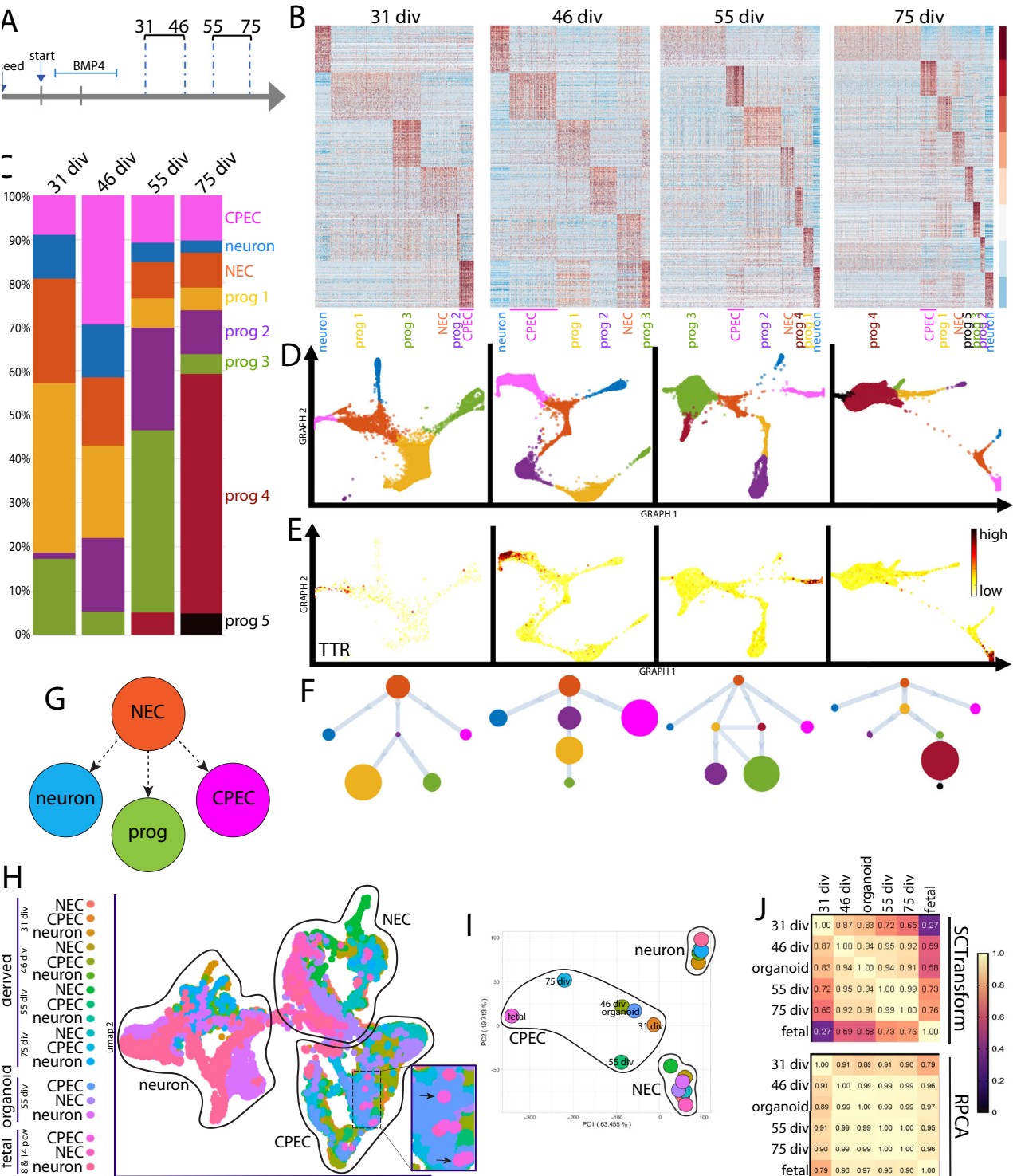

**Fig. 3 | Direct dCPEC origin from tripotent NECs and comparison to human organoid and fetal CPECs.** SoptSC analyses; see Supplementary Fig. 3D–F for Seurat analyses. **A** Schematic of the four derivation timepoints and their pairings for culturing and scRNA-seq processing. **B** Heatmaps of top-300 DEGs per cluster. Cell types and color code designated below with dCPECs in pink. Color scale of Z-score of DEGs. **C** Stacked bar graphs of the percentage of cells in each cluster across the datasets, color coded as in (**B**). dCPEC percentage is highest at 46 div, then decreases as neural progenitors continue to proliferate and diversify. **D** GRAPH charts, color coded as in B. At all timepoints, NECs (light brown) have direct branches to dCPECs (pink) and neurons (blue). **E** GRAPH feature plots highlighting dCPECs (TTR expression). **F, G** Unsupervised SoptSC pseudotemporal lineages (**F**) and general lineage model (**G**). NECs (light brown) have direct branches

to dCPECs (pink) and neurons (blue) at all timepoints. NEC relationships to progenitors ("prog") are more varied. **H** UMAP of dCPECs, NECs, and neurons aggregated with organoid and fetal CPECs (SCTransform-corrected). CPECs from all sources form a cluster. Fetal CPECs cluster with older dCPECs (inset). **I** PCA of clusters, color coded as in (**H**). The 75-div dCPEC cluster approaches the fetal CPEC cluster, while organoid CPECs display intermediate differentiation. **J** Pearson correlation tables (color scale ranges from r = 0 to r = 1) using two batch correction methods. With SCTransform (top), CPECs display more differences. The dCPECs progress in an orderly temporal fashion towards the organoid, then fetal CPECs. RPCA correction (bottom) highlights CPEC similarities regardless of source. See Supplementary Fig. 4C–D for complete correlation tables.

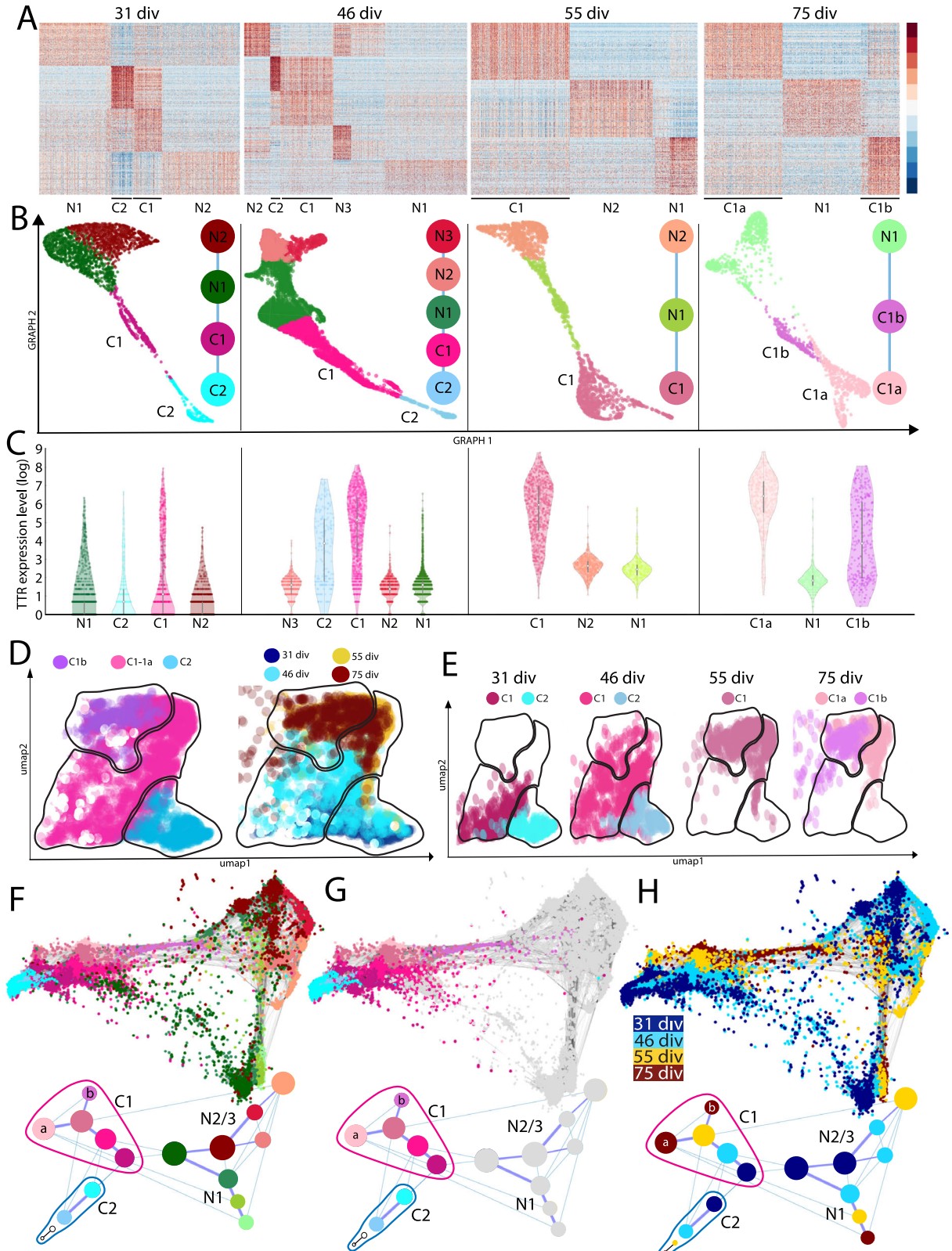

(Fig. 7F, G). Similarly, in the human tissues, CLDN5 and SC5D expression were positively correlated at 23 pcw, then became negatively correlated by 42 pcw (Fig. 7I–K; Supplementary Fig. 9I–K,M) and also appeared clustered (Fig. 7I, J; Supplementary Fig. 9J). Together with the scRNA-seq analyses, these studies indicate a C1a-C1b lineage bifurcation that occurs between 61–75 div in vitro and between 23–42 pcw in vivo.

## Discussion

In this report, we present an improved derivation of human CPECs, which differentiate with substantial cell autonomy and confirm several developmental principles from other species and systems[1,16] (Fig. 1; Supplementary Fig. 1). The human dCPECs also have dynamic multi-ciliated phenotypes that reciprocally interact with Aβ uptake (Fig. 2; Supplementary Fig. 2), while pseudotemporal, time series, and cell

**Fig. 4 | SoptSC and STITCH analyses across timepoints suggest a branching dCPEC lineage tree. A** Heatmaps of the top-300 DEGs for dCPECs ("C") and NECs ("N") suggest two dCPEC subtypes at 3 of the 4 timepoints, but only one cluster at 55 div. Color scale of Z-score of DEGs. **B** SoptSC GRAPH charts and lineage relationships coded by subtype. NECs are more similar to C1 than C2 cells. See Supplementary Fig. 5C for pseudotemporal color coding. **C** Violin plots (with median and quartiles) of TTR expression in the cells of each cluster across the datasets, which increases in dCPECs over time with C1 > C2 and C1a > C1b. Every dot represents a cell. **D** UMAPs of aggregated dCPECs, color coded and encircled for simplified subtype groupings (left) or timepoint (right). C2 cells (lower right) are mainly present early, but also at later timepoints, while C1 cells shift over time.

**E** UMAPs of dCPECs from individual timepoints, color coded for dCPEC subtype as in B. As C2 cells contract, C1 cells adopt a "hybrid" C1a-C1b profile at 55 div before specifying into C1a and C1b subtypes by 75 div. **F**–**H** STITCH dot plots (upper) and lineage trees (lower), color coded for all subtypes (**F**), dCPEC subtypes only (**G**), or timepoint (**H**). The dCPECs cluster towards the left of the roughly-triangular dot plots, with NECs at the other two corners. Each corner has cells from all four timepoints (**H**). C1 (pink-purple) and C2 cells (blues) belong to distinct lineages from the outset. The C2 lineage displays transience, while the C1 lineage evolves and diversifies. Small 55-div and 75-div C2 "clusters" were manually added to the lineage trees to indicate their presence in low numbers at these timepoints.

cycle scRNA-seq analyses reveal their direct origin from tripotent NECs as well as substantial CPEC subtype diversity (Figs. 3–5; Supplementary Figs. 3–7). Non-cycling type 1 (metabolic-energetic) and type 2 CPECs (ciliogenic) are present early in vitro and at midgestation in vivo (Fig. 6; Supplementary Fig. 8). By term, type 2 CPECs contract (Fig. 6; Supplementary Fig. 8) while type 1 CPECs bifurcate into type 1a (anabolic-secretory) and type 1b subtypes (catabolic-absorptive) (Fig. 7; Supplementary Fig. 9). Transcriptomic comparisons within this human NEC-CPEC lineage model (Fig. 7L) suggest the ontogenetic and phylogenetic emergence of type 2 and type 1b subtypes from a primordial NEC-type 1-type 1a lineage.

Molecular, cellular, functional, and transcriptomic findings established the identity and quality of dCPECs in the derivation protocol, which is more efficient, reproducible, scalable, and simple than our earlier methods[16,17]. In addition, dCPEC differentiation is relatively rapid and associated with phase-contrast milestones that are predictive and straightforward to assess (Fig. 1B; Supplementary Fig. 1). Published CPEC derivation methods[15,59,60] vary in their apicobasal topologies - i.e., whether apical or basolateral CPEC surfaces face the media. Using this protocol, dCPEC apical surfaces are media-facing (Fig. 1G; Supplementary Fig. 1J); culture media therefore corresponds to the CSF compartment, which can be advantageous. While BMP4 suffices for human dCPEC induction, others have coapplied BMP4 with a WNT signaling activator (CHIR 99021) to induce CPEC fate[15,59,60]. In our hands, BMP4 alone induced dCPEC WNT expression (Supplementary Fig. 6C), and CHIR 99021 provided no obvious additional benefit.

The early BMP4 requirement (Fig. 1C) for the direct NEC-CPEC lineage and its diversification (Figs. 3–7) support NECs being the CPEC-competent progenitor rather than radial glia[16,61], BMP4 sufficiency as a CPEC morphogen[16–18], the cell-intrinsic nature of NEC responses to BMP4[18,62], and the significant cell autonomy of CPEC differentiation[16,17], including subtype diversification. At a minimum, absence of non-neural cells in the cultures (Fig. 3; Supplementary Fig. 1D) indicates autonomy from mesenchymal, vascular, blood, and blood-borne influences, while the prominent 3D dCPEC folding (Fig. 1B, G; Supplementary Fig. 1G, K) is consistent with ChP morphogenesis being driven by CPECs rather than mesenchyme[63]. Early dCPEC differentiation (Fig. 3) and cell cycle exit (Figs. 1H, 5) also support CPECs being among the earliest differentiating cells in the human brain[11,40,64]. The presence of light and dark dCPECs (Supplementary Fig. 8I) suggests the autonomy of these ultrastructural and likely phasic, rather than developmental, differences among CPECs[8,9]. Our protocol also provides some insights, albeit limited, into neural induction. Absence of dCPECs upon BMP4 coapplication with NIM (0 div) is consistent with inhibition of BMP signaling being required to initiate neural induction, while the high efficiency of neural induction upon early BMP4 addition after NIM (1 div) indicates that complete BMP signaling inhibition may not be required to establish and maintain neural fate.

*BMP4* and *MSX1* gene expression by dCPECs[18,62] well after the cessation of exogenous BMP4 application (Supplementary Fig. 6C), the autocrine/paracrine BMP signaling by dCPECs predicted by CellChat[65]

(Supplementary Fig. 6D), and dCPEC island formation (Fig. 1; Supplementary Fig. 1) suggest that human CPEC induction is homogenetic ("like inducing like"), as postulated over a century ago[40] and as described for the roof plate[66,67] and telencephalic CPECs in mice[11,12]. BMPs, and BMP4 in particular, have been implicated in these homogenetic phenomena, which expand the space and time over which morphogens can act during development[68]. By 55 div, type 1 dCPECs have a mixed identity (Figs. 4B, 7D; Supplementary Fig. 5H) typical of proliferative intermediates in normal development and transdifferentiation[56] (Fig. 7D), but were non-cycling G0 cells (Figs. 1H, 5). Thus, unlike the irreversible G0 of most neurons[69], the G0 state of type 1 CPECs is presumably reversible.

The variably-high motility of early-stage dCPEC cilia (Fig. 6D; Supplementary Movies 1, 2) diminishes over time in vitro (Fig. 6E; Supplementary Movie 3). Zebrafish[49] and mice[50] also have motile CPEC cilia, and interestingly, this motility in mice diminishes postnatally[50]. For human dCPECs, decreased motility coincides with the loss of cilia clustering in vitro and in vivo (Fig. 2B–E; Supplementary Fig. 2B, C, E). We found no literature precedent for such losses, but for multiciliated cells, cilia motility is strongly associated with cilia clustering[70]. As in mice[50] and along with type 2 subtype contraction (see below), our motility and cilia clustering findings portend transience of CPEC cilia motility in humans.

Conversely, the enhancement of cilia clustering by exogenous $A\beta_{1-42}$ (Fig. 2I; Supplementary Fig. 2J) raises the possibility that Aβ uptake promotes cilia motility. To our knowledge, this study provides the initial demonstration of Aβ uptake by human CPECs, a potential Aβ clearance mechanism involved in Alzheimer's disease initiation and pathogenesis[71]. Preferential Aβ uptake into early endosomes (Supplementary Fig. 2H, I) surrounding clustered cilia (Fig. 2H) suggests that CPEC cilia serve as docking sites for uptake, as seen in other cell types[72,73]. While Aβ dispositions following this uptake remain to be fully described, these findings – together with the enrichment of Alzheimer's and other neurodegenerative disease pathways in all dCPEC subtypes (Figs. 6A, 7A) – raise the possibility that CPECs and the ChP contribute to the developmental origins of Alzheimer's disease[74,75] and potentially other neurologic and neuropsychiatric disorders that manifest later in life[76,77].

While dCPECs are generally multiciliated (Fig. 2A–E) and both type 1 and type 2 CPECs are enriched for ciliogenesis genes compared to non-dCPECs (Supplementary Fig. 8F), ciliogenesis gene enrichment further distinguishes type 2 cells (Fig. 6A–C). Type 2 CPECs correspond to the ciliogenesis-enriched CPECs described in developing mice[14] and human ChP organoids[15], with our study adding their early emergence, then contraction during prenatal human development (Figs. 4, 6) as well as the cilia dynamics just discussed. The prenatal contraction of type 2 CPECs draws comparisons to other transient cell types of the developing brain, such as Cajal-Retzius cells[78] and subplate neurons[79], and adds to the cell types with unique roles in early brain development. Our lineage analyses do not resolve whether NECs give rise to C1 and C2 cells directly or via a bipotential C1-C2 intermediate, or the symmetry/asymmetry of this event(s). Although pseudotemporal analyses

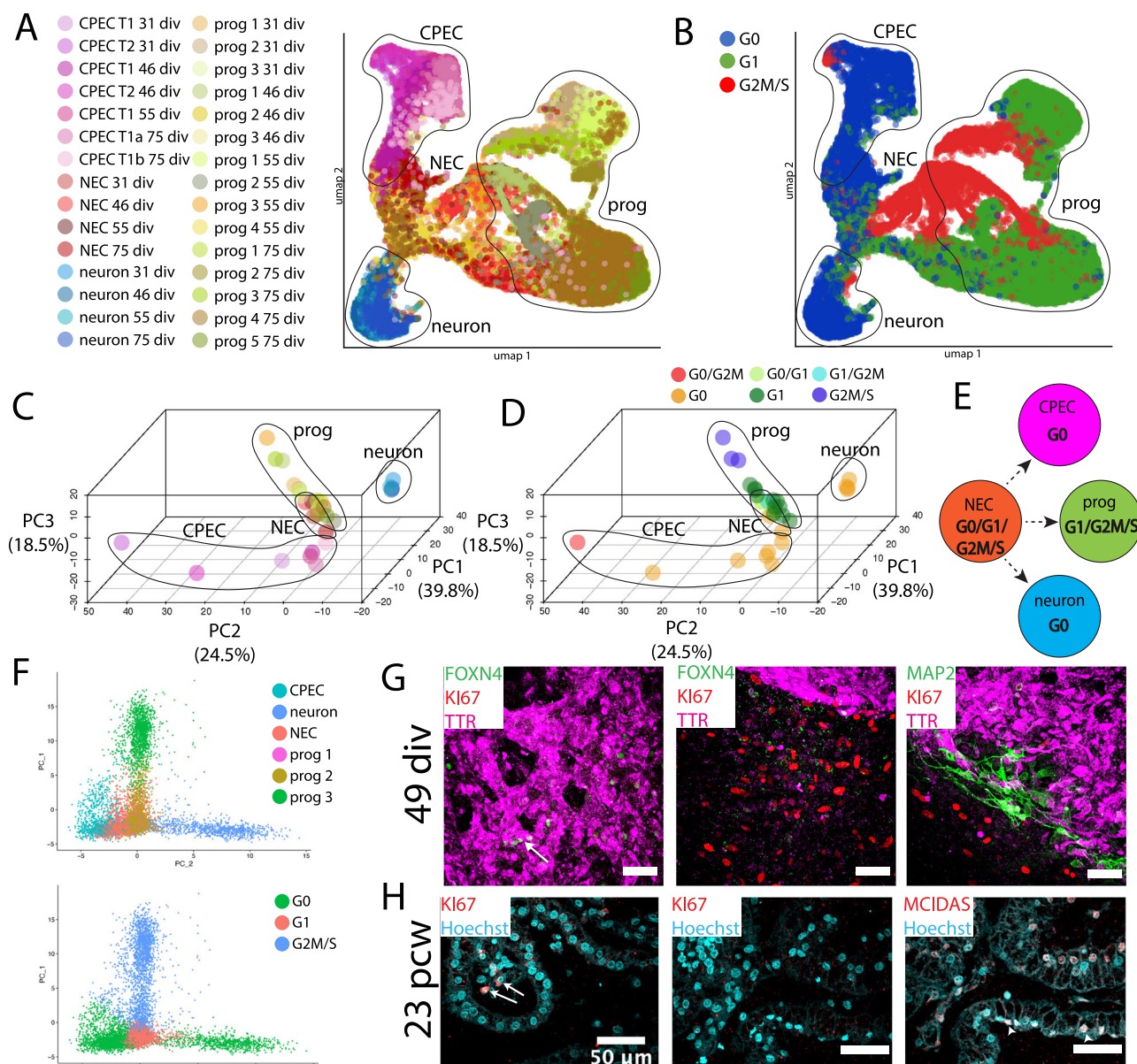

**Fig. 5 | Cell cycle analyses highlight NEC tripotency and reveal a distinctive 'neural G0' dCPEC signature.** Human 'neural G0' transcriptomic analysis[45] (**A**–**F**) with confocal maximal projections in derived cells (**G**) and 23 pcw tissue (**H**). **A, B** UMAP of all cells across timepoints, color coded for subtype and timepoint (**A**) or 'neural G0' phase (G0, G1, G2M/S), with major progeny classes encircled. The tripotent NEC progeny class structure is again evident (**A**). Across timepoints, dCPECs and neurons have predominant G0 signatures (blue), while NECs and progenitors are predominantly in G1 or G2M/S. **C, D** 3D 'neural G0' PCA plots of subtype clusters across timepoints, color coded for subtype as in A (**C**) or for predominant cell cycle phase(s) (**D**); see Supplementary Fig. 7A, B for details. The three branches of NEC progeny are evident from neural G0 analysis alone (**C**). While dCPECs and neurons are principally G0 cells (orange), neural G0 analysis also distinguishes dCPECs from neurons (**D**). **E** Lineage model for tripotent NECs (Fig. 3G) with predominant cell cycle signatures added. **F** 2D 'neural G0' PCA scatterplot of all cells from the earliest timepoint (31 div), color coded for cell type (upper) or cell cycle phase (lower). NEC tripotency and dCPEC-neuron distinction are evident by 31 div. **G** KI67 status of dCPECs at an intermediate timepoint in vitro (TTR·FOXN4·MAP2·KI67 ICC; 49 div). KI67 (red) labels cells outside of dCPEC islands (middle panel), but not dCPECs (purple) or neurons (green in right panel). FOXN4 + C2 cells (arrow in left panel) also lack KI67 expression. **H** KI67 status of CPECs at midgestation in vivo (MCIDAS·KI67 IHC; 23 pcw); middle and right panels are immediately adjacent. Nuclear KI67 is present in some stromal cells (arrow in left panel), but not in CPECs, including MCIDAS + C2 cells (arrowheads in right panel). See Fig. 6 for C2 marker studies using FOXN4 and MCIDAS.

did not reveal such an intermediate, C1 and C2 clusters were already present at the earliest stage profiled. Thus, additional studies will be needed to further resolve this initial CPEC lineage branch.

Along with the transience of cilia motility and clustering, contraction of ciliogenic type 2 cells suggests a conserved CPEC-to-ependymal transition in multiciliated ventricular cells that promote CSF flow and/or mixing. Nonami et al.[50] noted that mouse CPEC cilia motility—which peaks around birth before declining over the first two postnatal weeks—precedes the emergence of ependymal cells[80], whose cilia propel CSF[81], but do not mature until weeks after birth[82,83]. Likewise, in humans, ependymal multiciliation and maturation are largely postnatal and continue into adolescence[84], while CPEC differentiation[64], CPEC cilia clustering (Fig. 2B–E; Supplementary Fig. 2B, C, E), and type 2 cells (Figs. 4, 6) are evident much earlier *in utero*. The prominent "bloom" of telencephalic ChP in cynomolgus monkeys and early first-trimester humans[64,68,85] would further facilitate CPEC-mediated CSF flow and/or mixing before ependymal cells are able to.

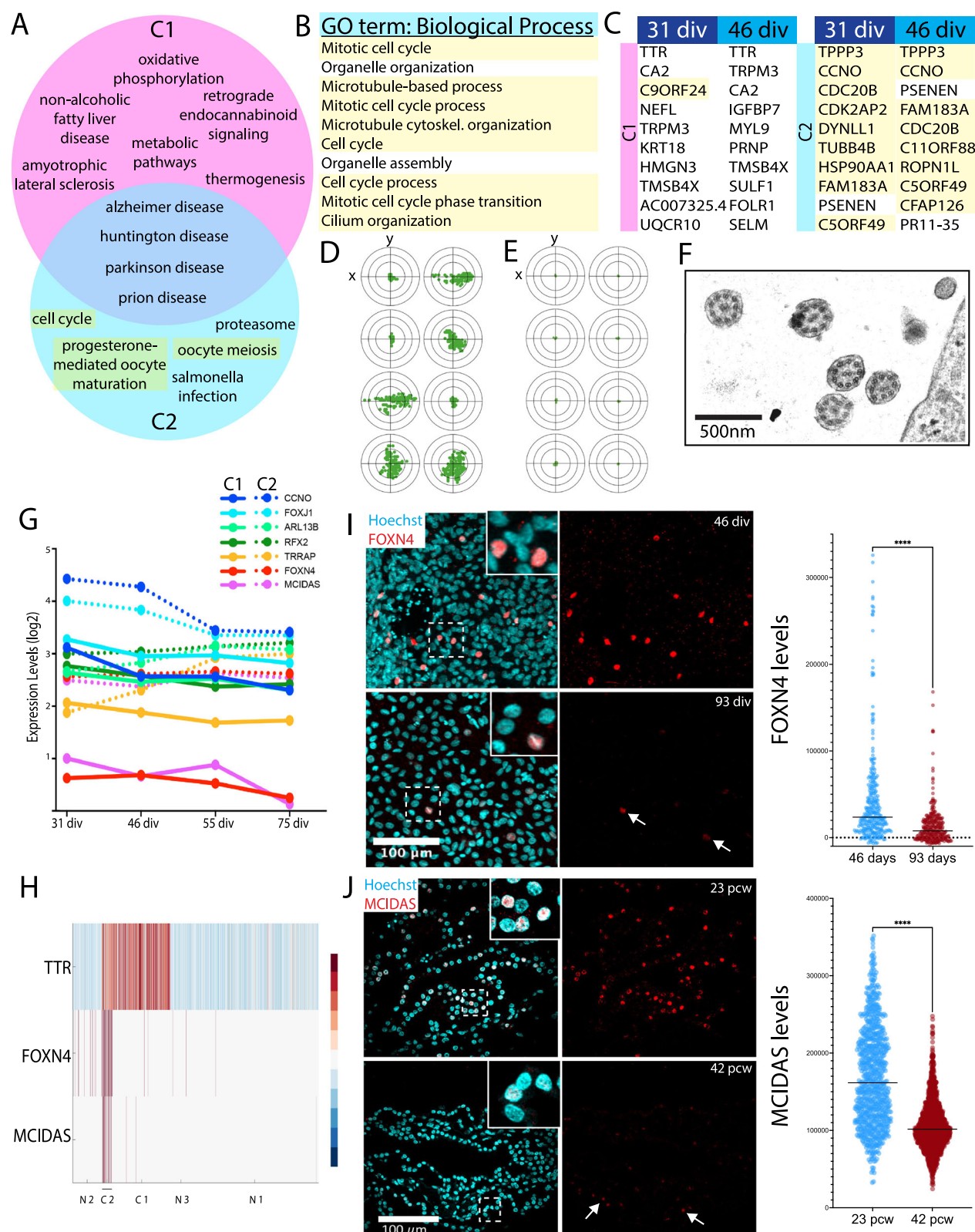

Type 1a-1b bifurcation does not occur until later in vitro or near term in vivo (Figs. 4, 7), and this late prenatal timing suggests preparation for birth as the ontogenetic driver. Other than conception and death, birth is the most significant physiologic transition in human ontogeny due to the abrupt separation and loss of maternal, placental, and amniotic fluid supports for fetal energy flux (energy substrate

delivery, usage, and waste clearance)[86,87]. Two well-known adaptations that support this dramatic transition in energy flux are the specification of surfactant-producing alveolar type II cells (AT2 or AEC2) in the lung and globin gene switching in the bone marrow. Both initiate during the late prenatal period with autonomy from the human birth process itself [88,89] and directly impact the fluids for energy flux (air and

**Fig. 6 | Early type 1 and type 2 CPECs, with later type 2 contraction. A**, **B** Venn diagram of all enriched KEGG pathways ($p < 0.05$ with Bonferroni correction) (**A**) and top-10 GO terms for C2 cells (**B**) based on top-300 DEGs of C1 and C2 clusters at 31 div. Microtubule/ciliogenesis-associated terms are highlighted. C1 (pink) is enriched for metabolic and energy-related pathways, while C2 (blue) is enriched for microtubule/ciliogenesis terms. C1 and C2 subtypes share enrichment for neuro-degenerative disease pathways. **C** Top-10 DEGs for C1 (left) and C2 subtypes (right) at 31 and 46 div, with microtubule/ciliogenesis-associated genes highlighted. C2 cells express ciliogenesis genes at particularly high levels. **D**, **E** Live-image cilia tracking at 42 div (**D**) and 110 div (**E**) of one cilia tip every 100 frames (0.5 s) over 50 s (100 dots). Variable motility patterns are observed at 42 div, but not at 110 div (target circles, 4.14 um in diameter). See also Supplementary Movies 1–3. **F** Trans-mission electron micrograph from a 40-div dCPEC culture. Five axonemes contain the central pair of singlet microtubules and "9 + 2" arrangement characteristic of motile cilia. **G** Gene expression levels ($\log_2$) of seven master regulators of ciliogenesis[52]. C2 cells (dashed lines) express higher levels than C1 cells (solid lines), particularly FOXN4 (red) and MCIDAS (purple). **H** Heatmap of NEC and dCPEC subtypes at 46 div. FOXN4 and MCIDAS distinguish C2 from C1 cells and NECs. Color scale of Z-score of DEGs. **I** C2 contraction in vitro (FOXN4 ICC maximal projections and violin plots with medians; 46 and 93 div). FOXN4 fractional posi-tivity and expression levels decrease between 46 and 93 div ($t$-test $p < 0.0001$****; $n = 300$ cells each), although C2 cells are detectable at 93 div (arrows). **J** C2 con-traction in vivo (MCIDAS IHC maximal projections and violin plots with medians; 23 and 42 pcw). Like FOXN4 in vitro, MCIDAS fractional positivity and expression levels decrease between 23 and 42 pcw ($t$-test ****$p < 0.0001$; 23 pcw $n = 799$ cells, 42 pcw $n = 1504$ cells), although C2 cells are detectable at 42 pcw (arrows). Source data provided as a Source Data file. See Statistics and Reproducibility section of Methods for additional information.

blood) in neonates and adults - i.e., transport of oxygen, glucose, other energy sources, carbon dioxide, heat, and other forms of metabolic waste to and from tissues and the body.

Notably, CPECs also produce and directly regulate a fluid (CSF), albeit one that selectively subserves the brain. Hence, type 1a and 1b CPECs likely evolved to meet the extrauterine energy flux demands of the human brain, including the replacement of *in utero* functions provided by the placenta, which shares many of the same top enri-ched pathways as type 1b cells (e.g., waste removal, immune reg-ulation, and steroid hormone production; Supplementary Fig. 9B)[90]. While AT2 cells and globin gene switching are conserved among mammals[91,92], type 1a and 1b CPECs have not been described pre-viously. As mentioned, the human fetal[36] and ChP organoid datasets[15] lack these subtypes (Supplementary Fig. 6F), but were sampled before type 1a-1b specification occurs. Regardless, while additional studies will be needed to define CPEC subtype phylogeny, *brain* phylogeny provides ample rationale for the ontogeny and phylogeny of unique CPEC subtypes in humans. Like AT2 cells and globin genes, fundamental CPEC roles center on energy flux and homeostasis needs, and these needs became massive in human brains. The absolute and relative expansion of hominin and human brains, especially their neocortices[93,94], includes an exponential relative burst in *Homo sapiens*[93,95] and associated changes in energy alloca-tion that were just as dramatic. Despite representing only 2% of body weight, the adult human brain uses 20–25% of the energy budget compared to 8–10% in non-human primates and 3–5% in other mammals[96], with the neonatal human brain expending a remarkable 60–65%[97].

While CPEC subtype evolution to support the increased energy flux demands of human brains has not been described previously, molecular and physiological system adaptations have been. Notably, many of these rely on CPECs for their implementation in the brain. These adaptations include the thyroid hormone (TH) system, which also co-evolved with human encephalization[98]; the basal metabolic rate[99] regulated by TH[100]; and the TH carrier and distributor system[101], including TTR[102,103], the most abundant CPEC product[13] whose expression is enriched in type 1 and type 1a CPECs (Fig. 4C). These and other evolutionary adaptations would align with the preterm specifi-cation of specialized CPEC subtypes to support the uniquely energy-demanding human brain.

## Methods

### Ethical regulation of research
These studies were carried out under the supervision and approval of local ethics committees (UCI IBC, IRB, and hSCRO). Postmortem ChP tissue studies were determined to be non-human subjects research by the UCI IRB. Research was conducted only after sufficient laboratory and safety training had been provided to all researchers.

### Pluripotent stem cell culture
H1 embryonic stem cells (ESCs; WiCell Research Institute, Madison, WI) and iPSCs (APOE3/3 and APOE4/4 isogenic pairs of UCI ADRC Line 6, Mount Sinai TCW line 1, Mount Sinai TCW line 2, and Coriell WTC-GFP) were cultured on growth factor-reduced Matrigel (Corning #354230) diluted in DMEM (Thermo Fisher #11995-065) and maintained in E8 media (Thermo Fisher #A1517001) at 37 °C in 5% $CO_2$ with daily full media changes. After reaching 80% confluence, PSCs were dissociated using ReLeSR following manufacturer protocol (StemCell Technolo-gies #05872), passaged using E8 with 10uM Y-27632 (ROCK inhibitor; StemRD #Y-025). Y-27632 was removed after 12–24 h. For quality controls, PSCs passed array CGH testing (Cell Line Genetics, Madison, WI) and pluripotency testing using a four-antibody kit (Thermo Fisher #A24881). Cell culture media were sterile filtered (Fisher Scientific #SCGP00525 and #0974063D), but no antibiotics or antimycotics were used per best practice guidelines[104]. Contamination was mon-itored by phase-contrast microscopy and mycoplasma detection kits (Thermo Fisher #M7006).

### CPEC derivation
Glass chamber slides (Millipore #PEZGS0816) were coated with growth factor-reduced Matrigel (0.085 mg/mL per chamber slide well). PSCs were dissociated, as above, and with trituration using a p1000 pipette tip to achieve clumps averaging 50 um in diameter. Cell solution was diluted to achieve 10 clumps/mm² density before being distributed evenly in the wells and incubated at 37 °C in 5% $CO_2$. ROCK inhibitor was removed after 24 h. Once PSC colonies reached ~150 um in dia-meter (usually 2–3 days post-seeding), PSC media was fully replaced with Neural Induction Media (NIM; Thermo Fisher #A1647801), which marked the start of derivation ("N0"). Twenty-four hours later ("N1") media was replaced fully with NIM with 10 ng/mL BMP4 (R&D Systems #314-BP-050). On day N5, media was replaced fully with CPEC media composed of DMEM/F12 (Thermo Fisher #11330-032), NEAA (Thermo Fisher #07980), N2 (Thermo Fisher #17502048), and Heparin (Stem-Cell Technologies #07980), supplemented with 10 ng/mL BMP4; day N5 became "C5". After 15 days of BMP4 exposure (day C16), cultures were given daily half-media changes with CPEC media without BMP4 for the remaining time in culture. Derivations with H1, UCI ADRC line 6 isogenic pair, and Coriell WTC-GFP isogenic pair were characterized with immunostaining and scRNA-seq while the other lines were con-firmed for CPEC derivation with immunostaining only.

### Optimization of derivation variables
Variables involved in CPEC differentiation involved seeding density, seeding clump size, Matrigel thickness, glass or plastic surface, BMP4 addition, BMP4 concentration, BMP4 duration, neural induction start, neural induction duration, neural induction method, maturation additives, and length of culture. Seeding densities tested ranged from 2–20 clumps/mm². Seeding clump sizes tested ranged from

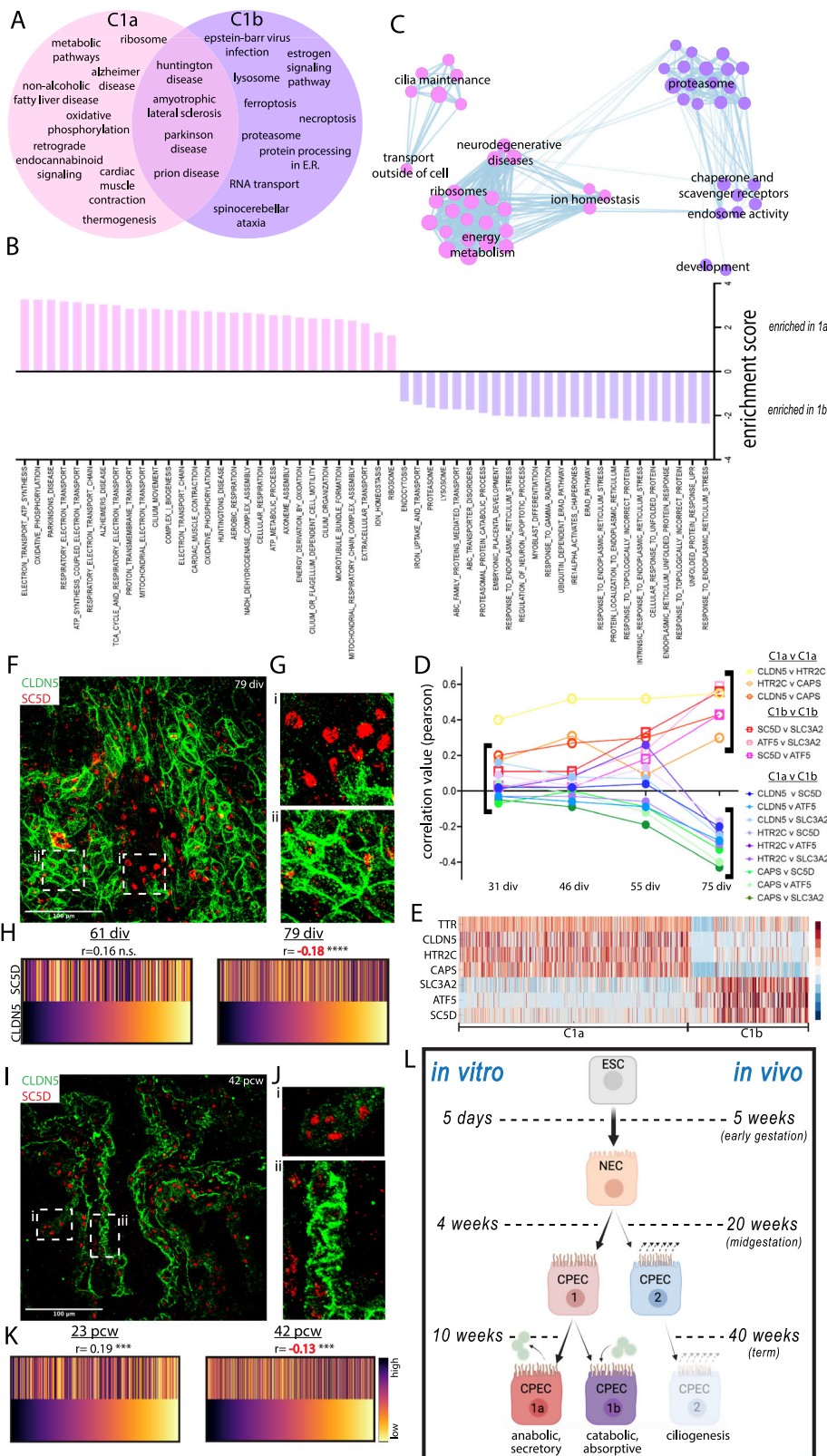

5–100 um/clump. Matrigel thicknesses tested ranged from 0.04–0.68 mg/mL. Plastic, glass coverslip, and glass slides were tested for culturing. BMP4 additions tested ranged from 1 day prior to the start of neural induction to 3 days post start of neural induction. BMP4 concentrations tested ranged from 5–20 ng/mL. BMP4 durations tested ranged from 5–35 days. Neural induction initiations tested ranged from 1 day post seeding to 5 days post seeding. Neural induction

durations tested ranged from 3–15 days. Neural induction method was compared between Thermo Fisher's product and StemCell Technologies' product. Maturation additives tested were with or without N2 supplementation, with or without cyclopamine, and with or without CHIR 99021. Length of cultures tested ranged from 10 to 100 days. These factors were tested one at a time initially and then in various combinations. In addition to phase-contrast morphology and TTR ICC,

**Fig. 7 | Late type 1a and type 1b CPECs with prenatal lineage summary. A** Venn diagram of significantly-enriched KEGG pathways (p < 0.05 with Bonferroni correction) based on top-300 DEGs of C1a and C1b cells at 75 div. C1b cells (purple) are distinctively enriched for catabolic processes, while C1a cells (pink) share many metabolic and energy-associated pathways enriched in 31-div C1 cells (Fig. 6A). See Supplementary Fig. 9A and Supplementary Data 1 for directed GSEA analyses of anabolic-secretory and catabolic-absorptive pathways. **B** Enriched GSEA terms for C1a (pink) and C1b cells (purple) using filtered genes (see Methods) expressed by >25% of cells. **C** Annotated GSEA theme plot illustrating C1a and C1b pathway enrichments. **D** Pearson correlations among (red-yellow) or between (green-violet) C1a and C1b marker gene pairs. Note the increase in some correlations at 55 div before their strong anti-correlations by 75 div. **E** 75 div heatmap of C1a and C1b marker genes used in panels **D**–**K** and Supplementary Fig. 9C–K. Color scale of Z-score of DEGs. **F**–**H** C1a-C1b specification in vitro (CLDN5-SC5D ICC maximal projections with magnified insets and rank-order heatmaps with Spearman correlations; 61 and 79 div). Unlike CLDN5-SC5D coexpression at 61 div (Supplementary Fig. 9F), discrete regions of CLDN5$^{hi}$-SC5D$^{lo}$ (C1a) and SC5D$^{hi}$-CLDN5$^{lo}$ (C1b) cells are evident by 79 div (**F**, **G**). A trend towards positive CLDN5-SC5D correlation at 61 div (p = 0.2$^{n.s.}$; n = 150 cells) becomes significantly negative by 79 div (p < 0.0001****; n = 584 cells) (**H**). **I**–**K** C1a-C1b specification in vivo (CLDN5-SC5D IHC maximal projections with magnified insets and rank-order heatmaps with Spearman correlations; 23 and 42 pcw). Similar to in vitro, discrete regions of CLDN5$^{hi}$-SC5D$^{lo}$ (C1a) and SC5D$^{hi}$-CLDN5$^{lo}$ (C1b) cells are evident by 42 pcw. At 23 pcw, CLDN5-SC5D correlation is significantly positive (p < 0.001***; n = 323 cells) before becoming significantly negative by 42 pcw (p < 0.001***; n = 812 cells). Color scale of fluorescence levels for CLDN5 and SC5D. **L** Summary of the CPEC lineage tree with in vitro and in vivo timepoints. Source data provided as a Source Data file. See Statistics and Reproducibility section of Methods for additional information.

CPEC differentiation efficiency was secondarily based on AQP1, CLDN1, AE2, and ARL13B ICC. For alternative neural induction (Supplementary Fig. 1C), SMADi Neural Induction kit (StemCell Technologies #08581) was applied 48 h after seeding fully dissociated H1 ESCs for 5 days. ESCs were dissociated using ReLeSR to obtain a single cell state, suspended in E8 + RI (10uM), diluted to 500,000 cells/mL, and then 1 mL of the solution was added to a single well of a 4-well chamber slide. Rock inhibitor was removed after 24 h (P1). Fourty-eight hours after initial seeding (P2 = N0), the cells were given SMADi neural induction media (StemCell Technologies #08581) for 5 days. 10 ng/mL BMP4 was added 48 h after kit application followed by daily half-media changes with CPEC media. After the 5 days of neural induction (N5 = C5), the media was fully replaced with CPEC + BMP4 media and after a total of 15 days exposure of BMP4 (C17), cultures were given daily half media changes with CPEC media without BMP4 for the remaining time in culture and yielded >90% CPECs.

### EdU (5-ethynyl-2'-deoxyuridine) incorporation
The Click-iT EdU Alexa Fluor 555 kit (Thermo Fisher #C10338) was used per manufacturer protocol. Cultures were washed 2x with CPEC media, then given a full media change with 40 uM EdU in CPEC media, followed by daily half-media changes with EdU-CPEC media. At termination, cultures were aspirated, washed 1x with DPBS, then fixed with 4% PFA (Fisher Scientific #50-980-487) for 15 min at room temperature (RT). Click-iT reactions were performed following manufacturer protocol prior to immunostaining, Hoechst counterstaining, and coverslipping.

### MitoTracker Orange (MTO) labeling
MTO dye (Thermo Fisher #M7510) was reconstituted in DMSO (Thermo Fisher #D12345) per manufacturer protocol and used with minimal light exposure. Biological triplicate wells were gently washed 2x with CPEC media, then given full media changes with 150 nM MTO in CPEC media or vehicle in CPEC media. After 20 min at 37 °C in 5% CO$_2$, media was aspirated, cultures washed once with DPBS, then fixed with 4% PFA prior to immunostaining.

### Aβ uptake
FAM-labeled Aβ$_{1-42}$ peptide (AnaSpec #AS-23526-01) was reconstituted following manufacturer protocol and used with minimal light exposure. Stocks were diluted to 500 nM-4 uM working solutions in CPEC media. Prior to treatment, cultures were gently washed 1x with DPBS, then CPEC-Aβ$_{1-42}$ media was added to triplicate wells and incubated at 37 °C in 5% CO$_2$ for 30 min-48 h. To address slide-to-slide variability, conditions were randomized across wells of different chamber slides. All conditions were stopped at the same time by aspiration, gentle washing 2x with DPBS, and fixation with 4% PFA for 20 min at RT.

### TTR ELISA with brefeldin A (BFA) inhibition
After 3x washes with warm CPEC media, 1 ml of CPEC media was reapplied to each well of a 4-well chamber slide, then 20 ul of conditioned CPEC media was longitudinally collected from individual wells over time (0–10 hrs starting with 1 ml, ending with ~820 ul in each well). To inhibit secretion, BFA (Invitrogen #B7450) or vehicle (DMSO) was added directly to wells. To test later for BFA reversibility, media was aspirated, then cells were washed 3x before replacing with CPEC media with or without BFA. Longitudinally collected media were assayed using a human TTR ELISA kit (AssayPro #EP3010-1) per manufacturer protocol and read on an HT microplate reader (BioTek Synergy). Background media-only values were subtracted, and concentrations were inferred from values normalized to the 2 h timepoint using standard curves measured in parallel.

### Immunostaining
Antibody information is provided in Table 1.

**ICC of adherent cultures.** Cultures were gently washed 2x with DPBS at RT, fixed with 4% PFA at RT for 20 min, then washed 3x with DPBS. Cultures were incubated in blocking solution (5% donkey serum (Jackson ImmunoResearch #017-000-121) in PBS + 0.03% Triton X-100 (MilliporeSigma #9036-19-5)) at RT for 1 h, followed by incubation in primary antibody solution (primary antibody diluted in 1% donkey serum in PBS + 0.03% Triton) at 4 C overnight. The next day, cultures were washed 3x followed by incubation in secondary antibody solution (secondary antibody diluted in 1% donkey serum in PBS + 0.03% Triton) at RT for 1 h in the dark, washed 3x, then incubated in Hoechst solution (2ug/ml in PBS) (Thermo Fisher #H3570) for 5 min at RT in the dark, then washed 3x. The chamber slide apparatus was then removed per manufacturer protocol and slides were mounted in Fluoromount-G (Southern Biotech #0100-01) and coverslipped (Thermo Fisher #102460).

**IHC of sectioned ChP.** Paraffin-embedded human ChP tissue, fixed in 10% formalin at RT, were sectioned at 5um thickness by histology service (Experimental Tissue Resource). Sections were heated at 65 °C for 30 min, deparaffinized in xylene (Sigma-Aldrich #214736), then rehydrated through a descending alcohol series. Antigen retrieval using sodium citrate pH 6 (MilliporeSigma #6132-04-3) for 20 min in a vegetable steamer at 100 °C was followed by 1x ddH20 wash for 10 min at RT. Hydrophobic Pap-Pen barriers (Thermo Fisher #R3777) were drawn before fluorescent immunostaining and coverslipping, as described above.

**IHC of ChP whole mounts.** Postmortem human ChP tissue from the lateral ventricle was obtained at autopsy, fixed and stored in 10% formalin at RT or 4 C. Under stereo dissecting microscope, multiple small pieces (<1 mm diameter) were randomly picked and placed in a mesh basket (Corning #CLS3478) in a 12-well plate. Free-floating immunostaining was performed as described above for sections using a rocker on low settings. Stained tissue pieces were placed on glass slides containing spacers (Invitrogen #S24735), then coverslipped.

**Table 1 | Information on antibodies used in this study**

| Primary Antibodies | | | | | | |
|---|---|---|---|---|---|---|
| Antibody | Vendor | Catalog # | Dilution ICC | Dilution IHC | Species | Localization |
| AE2 | S.C. BIOTECH | SC46710 | 1/250 | 1/200 | goat | membrane basolateral |
| AQP1 | EMD MILLIPORE | AB2219 | 1/1000 | 1/1000 | rabbit | membrane apical |
| ARL13B | PROTEINTECH | 17711-1-AP | 1/1000 | 1/1000 | rabbit | apical cilia |
| ATPB | ABCAM | AB14730 | 1/500 | NA | mouse | mitochondria |
| CLDN1 | INVITROGEN | 717800 | 1/500 | 1/50 | rabbit | membrane apical |
| eea1 | CELL SIGNALING | 2411S | 1/250 | NA | rabbit | early endosomes |
| Nestin | EMD MILLIPORE | ABD69 | 1/500 | NA | rabbit | interim. filaments |
| OTX2 | S.C. BIOTECH | SC133873 | 1/100 | NA | mouse | nuclear |
| SOX2 | INVITROGEN | PA1-094 | 1/300 | NA | rabbit | nuclear |
| TTR | ABCAM | AB9015 | 1/3000 | 1/3000 | sheep | secretory apparatus |
| ZO-1 | INVITROGEN | 339100 | 1/500 | 1/500 | mouse | membrane apical |
| FOXN4 | S.C. BIOTECH | sc-377166 | 1/500 | 1/50 | mouse | nuclear |
| MCIDAS | INVITROGEN | PA5-67092 | 1/500 | 1/50 | rabbit | nuclear |
| CLDN5 | INVITROGEN | 35-2500 | 1/200 | 1/50 | mouse | membrane apical |
| HTR2C | SINO BIOLOGICAL | 203151-T42 | 1/500 | 1/100 | rabbit | membrane |
| ATF5 | S.C. BIOTECH | sc-377168 | 1/500 | NA | mouse | nuclear |
| SC5D | INVITROGEN | PA5-67092 | 1/250 | 1/50 | rabbit | nuclear |
| Secondary Antibodies | | | | | | |
| Species | Vendor | Catelog # | Dilution ICC | Dilution IHC | Fluorophore | Channel |
| Donkey anti-mouse | Thermo Fisher | A31570 | 1/500 | 1/500 | 555 | red |
| Donkey anti-mouse | Thermo Fisher | A10038 | 1/500 | 1/500 | 680 | far red |
| Donkey anti-mouse | Thermo Fisher | A21202 | 1/500 | 1/500 | 488 | green |
| Donkey anti-sheep | Thermo Fisher | A11015 | 1/500 | 1/500 | 488 | green |
| Donkey anti-goat | Thermo Fisher | A11055 | 1/500 | 1/500 | 488 | green |
| Donkey anti-rabbit | Thermo Fisher | A31572 | 1/500 | 1/500 | 555 | red |
| Donkey anti-rabbit | Thermo Fisher | A32754 | 1/500 | 1/500 | 647 | far red |

### Image acquisition and modification

Phase contrast images of live cultures were acquired with an EVOS XL Core microscope (Advanced Microscopy Group). Fluorescent images were acquired with Olympus FV3000 confocal microscope, Nikon Eclipse E400 epifluorescent microscope with Nuance spectral deconvolution software, or Keyence BZ-X810 epifluorescent microscope for stitched images. All images were processed in ImageJ (v2.0.0). For all images compared qualitatively or quantitatively, identical image acquisition settings were used, and image adjustments in ImageJ were done in parallel and restricted to Brightness/Contrast.

### Transmission electron microscopy

Human dCPEC cultures were fixed with a 2% glutaraldehyde and 4% paraformaldehyde solution for 20 min at room temperature followed by cutting out the CPEC islands with a scalpel blade. Samples were kept in fixative for 5 days at 4 C followed by transfer to the UCI Experimental Pathology Core for processing and imaging.

### Image analysis and quantification

**MitoTracker Orange.** Confocal Z-stacks of cultures were converted to maximal projections. Membranous ZO1 signal was used to define TTR+ (CPEC) and TTR-negative (non-CPEC) ROIs, and total integrated MitoTracker Orange (MTO) signal was measured. Negative controls (no primary antibody) were imaged, then compartment-specific background signals were averaged and subtracted from experimental values. ROIs were drawn in blinded fashion prior to measuring integrated MTO values.

**Cilia, Aβ uptake, and early endosomes.** Confocal Z-stacks of cultures were converted into maximal projections. For cilia mass

measurements, membranous ZO1 was used to define cell ROIs, then total integrated ARL13B (cilia) and/or fluorescent $A\beta_{1-42}$ signals were measured. To assess cilia dispersion, outer contours of ARL13B signal were drawn and used as cilia ROIs. Cilia coverage was calculated by dividing cilia ROI area by cell ROI area. Negative controls were used for compartment-specific background subtraction, as above. For EEA1-Aβ colocalization measurements, the ImageJ co-loc2 tool was used. Due to inadequate signal using single cell ROIs, ROIs containing multiple TTR+ CPECs (5-10 cells) were drawn in blinded fashion before EEA1-Aβ colocalization measurements were taken.

**CPEC subtype markers.** Confocal Z-stacks of cultures and tissues were converted to maximal projections. For C2 markers (FOXN4 and MCIDAS), Hoechst channels were used to outline nuclear ROIs in ImageJ, which were then applied to the C2 marker channels to measure total fluorescent signal. For C1a and C1b markers, membranous CLDN5 signal was also used to outline cell ROIs. As above, ROIs were drawn in blinded fashion, and negative controls were used for compartment-specific background subtraction. Sample numbers are reported in Figure Legends; for the expanded in vitro and in vivo series in Supplementary Fig. 8C, D, all imaged and identifiable nuclei were scored.

**Videomicroscopy of motile cilia.** Live dCPEC cultures were imaged in an Olympus FV3000 confocal microscope equipped with CO2 and temperature regulation. Plane of focus was set to the apical surfaces of dCPEC islands. DIC movies were acquired using resonance scanning at 200 frames/second for 5 min intervals. Cilia tracking was performed in ImageJ with the Manual Tracking plugin. Using default settings, the tip coordinates of an in-focus cilium (one per dCPEC) were manually recorded every 100 frames, then plotted in two dimensions (X-Y graph).

## scRNA-seq data acquisition and quality controls

Multiple dissociation methods were tested per 10x Chromium guidelines. For each timepoint, two wells of a 4-well chamber slide were dissociated and pooled. The selected protocol involved dissociating with warm TrypLE (Thermo Fisher #12605010) for 25 min at 37 °C, resuspending in FACS buffer at RT (0.02% BSA in DPBS), then propidium iodine was added for 10 min (Thermo Fisher #P1304MP) before placing on ice. FACS for single live cells was performed by the UCI Flow Cytometry Core using a FACSAria II (BD Biosciences). Single cell capture with the 10x Chromium platform, cDNA library preparation, and cDNA QC analysis (Qubit dsDNA HS Assay kit (Life Technologies Q32851) and high sensitivity DNA chips (Agilent 5067-4626)) were performed by the UCI Genomics Research and Technology Hub (GRT Hub). Library generation was done with a V2 chemistry kit and an average capture of ~10,000 barcoded cells, per user guide (CG00052 Rev B). The cDNA libraries were sequenced using the HiSeq 4000 platform (Illumina) to achieve an average depth of 50,000 reads per cell and 2,500 genes per cell. FASTQ files were processed and mapped to the reference genome GRCh38 through Cell Ranger. scRNA-seq datasets were then bioinformatically filtered to remove low quality cells and reads as previously described[14,34–36]. Briefly, genes expressed in <10 cells were excluded, and cells with 1000–6000 genes/cell and ≤12.5% mitochondrial gene expression were included. The four time points were tested for their batch effect through two batch correction methods, RPCA and SCTransform with the Seurat platform. Datasets post quality control were combined with the merge function in Seurat while the time point of origin was retained in the metadata. RPCA batch correction was done with default settings of features to consider (2000) and plotted as a UMAP with color coding by time point of origin showing similar display of cells to the non-corrected. SCTransform batch correction was done with default settings of features to consider (all) and plotted as a UMAP with color coding by time point of origin showing similar display of cells to the non-corrected. Due to the similar grouping of cells in batch and non-batch corrected data, non-batch corrected data were used for further processing with the SoptSC platform.

## scRNA-seq data analysis

**SoptSC clustering and analysis.** These were performed as described[34]. We applied SoptSC (implemented in MATLAB R2019b) for clustering, marker gene identification, and lineage inference for each timepoint in the study. We selected 2000 highly variable genes (HVGs) as input to SoptSC for clustering and lineage estimation at each timepoint. Based on the number of zero eigenvalues and the largest eigengap of the clustering consensus matrix Laplacian, SoptSC provided a range of cluster-number estimates for each time-point. Cluster number per time point was done by finding the largest eigengap, picking the lower boundary of the gap, and generating a DEG heatmap to confirm the uniqueness of each cluster. For all time points the lower boundary was chosen, except for the 55 div set where the upper boundary of the eigengap showed cleaner clustering based on DEG heatmaps. We further determined the cluster number by verifying the expression of marker genes expressed by the clustered cells. For each specific time point, SoptSC infers lineage from cell-cell graph and abstract cluster-cluster graph using minimum spanning tree and was further validated by the biomarkers of the progenitor cell types.

**Seurat clustering.** Seurat[35] v4.1 analysis was performed per online Seurat tutorial. Each timepoint was processed independently. Highly variable features with standard deviations ≥2.5 were used for PCA generation. Resolution ranged from 0.1–0.3, and DEG heatmaps were used to validate clustering and identity similar to SoptSC.

**STITCH analysis.** For STITCH[39], highly variable genes were computed using Fano factors and ranking all genes by an above-Poisson noise statistic. Using default settings, top-2000 variable genes were filtered to include only genes whose single-cell transcript counts were minimally correlated (correlation coefficient >0.3) to at least one other variable gene. Gephi software was used for visualization.

**Neural G0 analysis.** ScRNA-seq datasets were merged, then Seurat was used initially for cell cycle assignment. Using the same Neural G0 classifier[45], either the mouse S and G2M gene lists from Seurat or the human G0, S, and G2M gene lists from Neural G0 were used. For Fig. 5D, cell clusters were assigned their predominant cell cycle phase unless a second phase exceeded 30%, in which case a "dual" phase was designated (see legend for Supplementary Fig. 7B).

**CellChat analysis.** For CellChat[65], preliminary imputation using Drimpute was performed to account for dropouts, and the imputation output was used as input. The CellChat vignette was followed to create various communication and summary plots (https://github.com/jinworks/CellChat).

**CPEC comparisons across datasets.** Raw reads and metadata of published datasets[15,36] were obtained from the UCSC Cell Browser. Datasets were processed through Seurat, as done by the originating groups, using the same quality control thresholds used for dCPECs. Cell types of interest (CPECs, neurons, NECs) were merged across datasets, followed by batch correction with either RPCA[37] or SCTransform[38] before clustering, Pearson correlations, and PCAs were performed in Seurat.

**Gene enrichment and pathway analyses.** Differential (top 300) and highly-expressed genes (top 10%) were used as input for pathway enrichment studies, then TopGO, KEGGprofile, and ReactomePA packages returned terms that reached statistical significance ($p < 0.05$ with Bonferroni correction). GSEA enrichment analysis compared all genes expressed in ≥25% of cells, also returning terms with $p < 0.05$ with Bonferroni correction. Heatmaps of manually-curated gene lists for specific CPEC subtype functions (Supplementary Fig. 9A, B) were generated after mining the GSEA human molecular signature database (https://www.gsea-msigdb.org/gsea/msigdb).

**Correlation and aggregate analysis.** C1a-C1b gene coexpression and divergence were explored by correlating the expression of every pairwise combination (15 in total) of the top 3 C1a and C1b DEGs across type 1 CPECs only at every timepoint with the FeatureScatter function to yield Pearson correlation values. Additionally, evidence for early emergence of late subtypes (C1a and C1b) and persistence of early subtypes (C1 and C2) was explored by aggregating the CPEC and NEC subtypes across the four timepoints and subjecting them to clustering, which generated 6 clusters supported by DEG heatmap. UMAP graphs of the aggregate dataset were color coded and/or split by timepoint or cluster of origin. See Supplementary Information for additional details.

## Statistics and reproducibility

Specific statistical tests for each experiment are mentioned in Figure Legends and Supplementary Fig. Legends. All $t$-tests were unpaired and two-tailed, while all ANOVAs were one-way. Both Pearson and Spearman correlation tests were two-tailed. Pearson correlations were used to compare transcriptome datasets and fluorescent levels. For rank-order analyses, Spearman correlations were used. Quantification of fluorescent signal in postmortem human tissue used multiple tissue sections across multiple glass slides. When possible, samples from different patients were used. Qualitative assessment of fluorescent signal also involved multiple tissue sections across slides per patient and images were representative across sections. Quantitative studies of dCPECs, apart from the TTR ELISA and MTO treatment, were performed with multiple chamber slides in at least three independent CPEC derivations. MTO studies were performed in two independent

derivations. TTR ELISA was performed once but involved longitudinal sampling across multiple slides and wells. Movies of cilia motility were performed on cells derived from the same initial plating and are representative of movies imaged from multiple wells and chamber slides. For studies tracking changes across the age of dCPECs, time-points were from the same initial plating. Immunostaining and imaging were performed once all replicates were generated. Quantification involved blinding using three different methods. Multiple users were employed who were blinded to the ages, conditions, and marker identity of the fluorescent and DIC images. While measurements were taken, users were blinded to other fluorescent channels as well.

## Graphing and AI attestation

GraphPad Prism8 was used for graphing and statistical analysis of the ELISA, fluorescent measurements, and transcriptome correlations. Custom illustrations (Fig. 7L) were created in BioRender. Masters, H. (2025) https://BioRender.com/cpj4ynj. All studies were conducted without the use of deep learning algorithms or platforms. Additionally, no form of AI was used for data analysis or writing this report.

## Inclusion and ethics

This study involved various local and regional researchers across multiple career stages. The collaborators in this study qualify for authorship as outlined by Nature Communications and are listed as authors. Their involvement and contribution were discussed prior to the start of the research. This work is locally relevant as determined by all collaborating researchers. The research was neither restricted by the setting of the researchers nor resulted in stigmatization, incrimination, discrimination, or personal risk to participants. Local and regional research relevant to our study was addressed properly in the citations and the report.

## Reporting summary

Further information on research design is available in the Nature Portfolio Reporting Summary linked to this article.

## Data availability

The single cell sequencing data generated in this study have been deposited in the NCBI GEO database under accession code GSE296691. All codes used in this study were publicly available through Bioconductor [https://www.bioconductor.org/] or through the institutions that published the platforms Seurat [https://satijalab.org/seurat/articles/pbmc3k_tutorial][35], STITCH[39] [https://github.com/wagnerde/STITCH], and SoptSC[34] [https://github.com/WangShuxiong/SoptSC?tab=readme-ov-file]. Source data are provided with this paper.

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

## Acknowledgements

We thank the UCI Genomics Research and Technology Hub (GRT Hub) for assistance with scRNA-seq; the Sue and Bill Gross Stem Cell Research Center for access to FACS, core equipment, and stem cell culture support; Julia TCW and Mt. Sinai, the ADRC iPSC Core, and Coriell Institute for iPSCs; the UCI Experimental Tissue Resource and autopsy service of the Department of Pathology & Laboratory Medicine for access to postmortem human tissues, histology, and transmission electron microscopy services; and members of the Monuki lab for support and feedback on the work and manuscript. This work was supported by NIH R21 AG064640, NIH R21 MH109036, and UCI SoM-SoBS pilot award (E.S.M.); NIH R25 GM055246, NIH T32 NS082174, and Rose Hills Foundation (H.M.); NSF DMS1763272 and Simons Foundation 594598 (QN); and CIRM EDUC2-08383 (VE, OGJ).

## Author contributions

H.M., B.A.J., and E.S.M. conceived the study. H.M., C.T., B.A.J., and E.S.M. contributed the theoretical basis for protocol optimizations, while H.M. and C.T. performed the optimizations. H.M. and Q.Nguyen acquired the scRNA-seq data. H.M., S.W., Y.S., M.K.K., N.K., K.K., Q.Nie, and E.S.M. contributed to scRNA-seq analysis. H.M., P.S.R.F., B.T., and C.M. contributed to fluorescent imaging and quantification. H.M., V.E., B.A.J., and E.S.M. contributed to statistical analysis. H.M., M.N., and O.G.J. contributed to cell culture treatments. H.M., B.A.J., and E.S.M. contributed to experimental planning and data analysis. H.M. performed the live videomicroscopy and analyzed the TEM photomicrographs, while H.M. and E.S.M. wrote and revised the manuscript. All authors approved the final manuscript.

## Competing interests

The authors declare no competing interests.
