## [Transparent Peer Review file · Nature Communications]

Sequential emergence and contraction of epithelial subtypes in the prenatal human choroid plexus revealed by a stem cell model

Corresponding Author: Dr Edwin Monuki

Version 0:

Reviewer comments:

Reviewer #1

(Remarks to the Author)

Masters et al. have established a novel protocol to differentiate human pluripotent stem cells (hPSC) into choroid plexus epithelial cells (CPECs). CPECs transcriptionally resemble fetal CPECs and are divided into two different subtypes (metabolic vs secretory). Importantly, CPECs diversification over time was compared to human fetal tissues.

While the functions of the choroid plexus are mostly performed by CPECs, little is known about those cells. This paper provides a great resource for generating different subtypes of human CPECs and it sheds some light on their developmental origin.

Overall the manuscript is novel and relevant for multiple fields, ranging from stem cell and neurodevelopment to neurodegeneration.

Here are a few points, that will strengthen the manuscript:

1. The authors nicely showed the hPSC-derived CPECs transcriptionally resemble fetal CPECs. It would be important to compare their transcriptomics profile to other brain and non-brain endothelial subtypes. So far, it has been challenging to generate brain endothelial cells (1) that show the correct expression of proteins known to be expressed by primary brain endothelial cells. Beyond Claudin5, shown in Figure 1 it would be interesting to show other markers, e.g. PECAM1+, CDH5+, EPCAM-, low junctional electrical resistance, etc.. It would be interesting to check those hallmarks in CPECs and eventually add a paragraph in the "Discussion" to discuss any eventual limitations of the system.
2. The number of batches used for scRNA-seq should be highlighted.
3. To prove the consistency and efficiency of the protocol, it would be important to show how it works in several hESC and iPSC lines. The authors already tested several iPSC lines (Figure S1), but it was not clear how many. Please add the name of these lines directly in the figure or figure legends. Was only one hESC line (H1) tested? At least a few hESC and iPSC lines should be tested.
4. Is it possible to isolate a pure culture of CPEC by, for example, FACS/MACs? It would be helpful for readers who might be interested in using those cells for follow-up studies.
5. Do you think that C2 cells can give rise to C1 cells at later time points, (55 div and 75 div) or that the culture simply does not support their survival/expansion? Does this explain the loss of FOXN4 and MCIDAS over time?

Minor points:

6. Figure 3F would be more helpful with labels directly on the figure.
7. Scale bars in Figure 5 don't have the same style and some are missing in Figure 5H
8. In Figure 5G-H it would be helpful to add the distinction "in vitro" vs "fetal" samples.
9. The figures' resolution is not always optimal. It's probably because of the reduction in the file size, but please check if this is correct.

Reviewer #2

(Remarks to the Author)

This manuscript by Masters et al., reports a new understanding of human choroid plexus development made possible by technical improvements made in the protocol used for generating derived CPEC cells. They carry out single cell transcriptomics in vitro and identify a CPEC lineage tree with several important observations. They determine that CPEC differentiate from neuroepithelial precursor cells, not radial glial cells. They further identified two bifurcations in the CPEC lineage. Type 1 cells (metabolic type), which subsequently further differentiated into anabolic-secretory (Tube 1a) and catabolic absorptive (tube 1b) cells. Type 2 cells, representing ciliogenic cells, contracted with time. They confirm a key function of CPECs to take up amyloid beta. The cell lineages and expression patterns were further validated, in many cases with direct comparison to human tissues. The work represents an important advancement for the field by providing a new protocol that improves efficiency, consistency, and scale, facilitating and helping to standardize in vitro studies for this important epithelium across laboratories. The confirmation of findings in human tissue specimens further strengthens the findings. The study represents an important advance to the field and will be of interest to the readership of Nature Communications.

While the manuscript already provides an important advance for the field, several additional suggestions would strengthen the study:

1. The authors clearly demonstrate the presence of cilia – is it possible to observe ciliary movement and formally demonstrate whether ciliary motility changes with advanced maturation of these cells in this experimental system, using for example light microscope or DIC optics?
2. The amyloid beta uptake nicely demonstrates the absorptive functions of the epithelial cells. Can the authors comment on what happens to the AB once taken up – what puncta is it retained in? Is it degraded by lysosomal degradation, or it is retained in the cultured cells for longer periods of time?
3. Do the authors see evidence of apocrine secretion (e.g., aposomes?) in cultured cells? HTR2C expression is used to mark epithelial cells. It would be interesting to know if the presently cultured cells could be triggered to undergo apocrine secretion for example following treatment with 5HT2CR agonist WAY-161503 (see PMID: 32961128 and 38260341).
4. Do the authors see evidence of light / dark cells?
5. It is interesting that CLDN5 emerges as a strong candidate in this study whereas Cldn5 expression in mouse choroid plexus epithelium seems rather low, with strongest expression in some of the major arteries of the choroid plexus (see PMID: 33932339). If the authors agree, it could be helpful to the field to mention that CLDN5 expression (or relative expression across the tissue) may vary based on species.
6. It would be helpful to include a summary schematic that depicts the findings of the study.

Reviewer #3

(Remarks to the Author)

This article showed lineage of human CPECs during differentiation from human ES cells and how the lineage correlate with in vivo choroid plexus development using stem cell culture model and transcriptome analysis. The content is very interesting. The data are analyzed by multiple ways, and the manuscript is well written. I have some concerns as follows;

Major points;

1. The authors showed the lineage of dCPECs differentiation and made comparison with in vivo development. However, dCPECs originate from ES cells and in vivo choroid plexus originate from roof plate. Thus, it is better to show the early phase correlation of dCPEC and in vivo choroid plexus. Besides, I am curious about the regionalization of NEC, C1 and C2, to C1a and C1b lineage separation. Does the lineage separation happen in salt and pepper pattern or localized pattern(or another way)? I think the authors can make these points clearer.
2. Regarding Figure2 B and E, the apicobasal polarity of the epithelium looks not clear. Fig2A showed clear staining of ARL3B in apical side, but in Fig2B/E, ARL3B looks as aggregates but does it mean the epithelium forms rosette-like structure? The authors can show this point.
3. Fig3C shows dCPEC is derived from NEC, and Fig5D shows CPEC and NEC colocalize in both early and later culture period. Is this mean NEC produce CPEC in asymmetrical cell division manner? Besides, in images of Fig1-2, almost all of epithelium looks to express TTR but how about the expression of NEC markers. Do they express NEC markers as well, or authors focused only in TTR expression region? The authors can show more equivalent image for the lineage specification for better understanding of readers.

Minor points;

1. Line147: FigS2G is a typo of FigS2H?
2. Line230-: The authors discussed that the higher rate of induction of lateral ventricle type of ChP is consistent with “default” model, but I think this can be derived from the effect of BMP treatment because 4V type of ChP development process need Shh signal but the LV type need BMP signals. The authors can make discussion with this aspect.
3. In general, it is thought that the inhibition of SMAD signals induce NEC from pluripotent stem cells, but the authors added

BMP4 from very early phase of NEC induction and some may think this is inconsistent with dual SMAD protocol. There are other protocols to induce neural fate by addition of BMPs from early phase of induction (Watanabe M et al. Stem cell reports 2022, Kuwahara A et al. Nature commn 2015), but is there some relationship between these articles with the author's result regarding the mechanism to induce neural fate via BMP signals?

Version 1:

Reviewer comments:

Reviewer #1

(Remarks to the Author)

The authors have addressed my concerns and I don't have any additional comments. Nice work !

Reviewer #2

(Remarks to the Author)

This is very nicely revised manuscript that addresses prior questions with new experiments, helpful explanations, and clarifications. Congratulations to the authors!

Reviewer #3

(Remarks to the Author)

The authors responded well to my major and minor points.

We thank the reviewers for their positive feedback, questions, and critiques that have strengthened this manuscript. Point-by-point responses are provided below in blue. Significant additions and revisions of text and figures (but not minor typos, rearrangements, or corrections) are also designated by blue font in the revised manuscript; corresponding line locations of these revisions are also provided below to simplify review. Lastly, we briefly describe additional time-series data (limited and supplemental only) that further support key conclusions of this manuscript:

Reviewer #1

“1. The authors nicely showed the hPSC-derived CPECs transcriptionally resemble fetal CPECs. It would be important to compare their transcriptomics profile to other brain and non-brain endothelial subtypes. So far, it has been challenging to generate brain endothelial cells (1) that show the correct expression of proteins known to be expressed by primary brain endothelial cells. Beyond Claudin5, shown in Figure 1 it would be interesting to show other markers, e.g. PECAM1+, CDH5+, EPCAM-, low junctional electrical resistance, etc.. It would be interesting to check those hallmarks in CEPCs and eventually add a paragraph in the “Discussion” to discuss any eventual limitations of the system.”

In addition to Figure 1, Figure 7 shows that CLDN5 is expressed by derived and primary CPECs and serves as a C1a subtype DEG and protein. Our derivations also lacked endothelial or non-neural cell types for internal comparisons, but based on this critique, we looked for PECAM1 and CDH5, as well as other positively-expressed brain endothelial genes (CD34, CD160, ACE), but did not detect these in the four scRNA-seq datasets (data now shown). Text has been revised to clarify the absence of non-neural cell types for direct comparisons (**line 172**) as well as text and references about CLDN5 expression in human and zebrafish CPECs (**lines 308, 1310 and 1313; references 101 and 102**). Reviewer #2 also raised a point about the potential species differences in CLDN5 expression, which we address below.

“2. The number of batches used for scRNA-seq should be highlighted.”

For each of the four scRNA-seq runs, cells were pooled from two wells of 4-well chamber slides that met phase-contrast quality controls. This has been added to Methods (**line 628**).

“3. To prove the consistency and efficiency of the protocol, it would be important to show how it works in several hESC and iPSC lines. The authors already tested several iPSC lines (Figure S1), but it was not clear how many. Please add the name of these lines directly in the figure or figure legends. Was only one hESC line (H1) tested? At least a few hESC and iPSC lines should be tested.”

We appreciate the catching of this omission. In addition to H1 ESCs, dCPECs have now been successfully derived from eight iPSC lines whose identifying names, sources, and confirmatory assays have been added to Methods (**lines 478 and 507**) and Acknowledgements (**line 460**). Images and text for one iPSC line (UCI ADRC6) have been retained in the original Figure S1 and accompanying figure legend.

“4. Is it possible to isolate a pure culture of CPEC by, for example, FACS/MACs? It would be helpful for readers who might be interested in using those cells for follow-up studies.”

We fully agree about the potential utility of a purification method for human dCPECs and are actively pursuing FACS, MACS, and other approaches. Unfortunately, we do not yet have a reproducible method to report, but given the potential interest, we will plan to do so via preprint and a small methods paper upon developing and validating such a method.

“5. Do you think that C2 cells can give rise to C1 cells at later time points, (55 div and 75 div) or that the culture simply does not support their survival/expansion? Does this explain the loss of FOXN4 and MCIDAS over time?”

The bioinformatic and ICC analyses suggest that C2 fate is “terminal” – C2 cells were present at all derivation stages, but contracted in number over time rather than bifurcating or differentiating further. Importantly, the parallel findings from human tissues suggests intrinsic (genetic) programming – rather than cell culture artefact – as the basis for C2 contraction, which in turn would account for the FOXN4 and MCIDAS reductions over time *in vitro* and *in vivo*. Further reflections also led us to analogous early-appearing, yet contracting cell types in brain development such as Cajal-Retzius cells and subplate neurons. Text and references have been added to further highlight the intrinsic programming of CPEC subtype diversification (**lines 347 and 350**) and the analogies to other transient brain cell types (**lines 399, 1316, and 1318; references 103 and 104**).

Minor points:

“6. Figure 3F would be more helpful with labels directly on the figure.”

Our attempts to achieve an aesthetic result with additional labels were unsuccessful. As an alternative, we better aligned panels 3F and 3G (**line 805**), which utilize the same color coding, and the accompanying figure legend (**line 813**).

“7. Scale bars in Figure 5 don’t have the same style and some are missing in Figure 5H”

Done.

“8. In Figure 5G-H it would be helpful to add the distinction “in vitro” vs “fetal” samples.”

Done. Timepoint labels have also been added.

“9. The figures’ resolution is not always optimal. It’s probably because of the reduction in the file size, but please check if this is correct.”

Images files were re-reviewed and, in some cases, increased in size to improve resolution.

Reviewer #2

“Q1. The authors clearly demonstrate the presence of cilia – is it possible to observe ciliary movement and formally demonstrate whether ciliary motility changes with advanced maturation of these cells in this experimental system, using for example light microscope or DIC optics?”

We thank the reviewer for asking this question. Indeed, we were able to visualize motile cilia on the derived CPECs by DIC optics as well as transmission electron microscopy (TEM). Supplemental videos S1 and S2 as well as new panels and legends for Figures 6 and S8 have been added. Correspondingly, text was added to Results (**line 266**), Discussion (**line 375**), Methods (**lines 588 and 619**), Contributions (**line 750**), Figure 6 (**lines 865 and 874**) and Figure S8 (**lines 1030 and 1046**). “Motile” or “motility” were added as conclusions to the Abstract (**line 31**) and Introduction (**line 76**). One reference has also been added (**line 1308; reference 100**).

“Q2. The amyloid beta uptake nicely demonstrates the absorptive functions of the epithelial cells. Can the authors comment on what happens to the AB once taken up – what puncta is it retained in? Is it degraded by lysosomal degradation, or it is retained in the cultured cells for longer periods of time?”

With minor text revision (**line 147**), we emphasized A β -EEA1 localization to early endosomes (EEA1+) and the decreasing A β -EEA1 colocalization over time, suggesting further intracellular trafficking of A β . Beyond these, however, we have no additional insights to offer, which we also clarified in Discussion (**line 387**).

“Q3. Do the authors see evidence of apocrine secretion (e.g., aposomes?) in cultured cells? HTR2C expression is used to mark epithelial cells. It would be interesting to know if the presently cultured cells could be triggered to undergo apocrine secretion for example following treatment with 5HT2CR agonist WAY-161503 (see PMID: 32961128 and 38260341).”

We thank the reviewer for this interesting question, which we are excited to explore in the future, but have not yet systematically examined.

“Q4. Do the authors see evidence of light / dark cells?”

The TEM studies for motile cilia also yielded ultrastructural evidence for light and dark dCPECs, which have been added to Figure S8 and its legend (**lines 1030 and 1046**) and briefly in Discussion (**line 356**).

“Q5. It is interesting that CLDN5 emerges as a strong candidate in this study whereas Cldn5 expression in mouse choroid plexus epithelium seems rather low, with strongest expression in some of the major arteries of the choroid plexus (see PMID: 33932339). If the authors agree, it could be helpful to the field to mention that CLDN5 expression (or relative expression across the tissue) may vary based on species.”

Agreed. Text on potential species differences regarding CLDN5 has been added (**line 308**).

“Q6. It would be helpful to include a summary schematic that depicts the findings of the study.”

A summary schematic had been provided as a figure panel (Fig. 7L) rather than a separate figure and was admittedly obscure. We enlarged and more boldly outlined the schematic to highlight further (**line 889**).

Reviewer #3

“Q1. The authors showed the lineage of dCPECs differentiation and made comparison with in vivo development. However, dCPECs originate from ES cells and in vivo choroid plexus originate from roof plate. Thus, it is better to show the early phase correlation of dCPEC and in vivo choroid plexus. Besides, I am curious about the regionalization of NEC, C1 and C2, to C1a and C1b lineage separation. Does the lineage separation happen in salt and pepper pattern or localized pattern (or another way)? I think the authors can make these points clearer.”

As correctly pointed out, midline CPECs differentiate from the roof plate *in vivo*, while bilateral CPECs in the telencephalon are induced from telencephalic neuroepithelium adjacent to the roof plate³⁸⁻⁴⁰. However, as previously observed for mouse dCPECs¹⁴, human dCPECs originate directly from NECs rather than ESCs (Figs. 3-5). Also, since the roof plate is neuroepithelial in origin, these observations are coherent. Nonetheless, we edited a sentence that could be misinterpreted to suggest direct CPEC origination from ESCs (**line 102**) and further emphasized the direct NEC-CPEC relationship (**line 347**). In addition, we agree with this reviewer that more “earlier phase correlation” would be beneficial, since current data do not fully resolve whether NECs give rise to C1 and C2 cells directly or via a C1-C2 intermediate. Text about this limitation has been added (**line 402**). Regarding the separate point about the apparent “salt and pepper” spatial distributions, we edited the text to convey this subjective, yet accurate impression (**line 280**).

“Q2. Regarding Figure 2 B and E, the apicobasal polarity of the epithelium looks not clear. Fig 2A showed clear staining of ARL3B in apical side, but in Fig 2B/E, ARL3B looks as aggregates but does it mean the epithelium forms rosette-like structure? The authors can show this point.”

Figs. 2B and 2E are *en face* views. In contrast to the *in profile* (orthogonal) views mentioned, apicobasal polarity cannot be established. We added text to the legend for Figure 2 to clarify (**line 779**).

“Q3. Fig3C shows dCPEC is derived from NEC, and FigS5D shows CPEC and NEC colocalize in both early and later culture period. Is this mean NEC produce CPEC in asymmetrical cell division manner? Besides, in images of Fig1-2, almost all of epithelium looks to express TTR but how about the expression of NEC markers. Do they express NEC markers as well, or authors focused only in TTR expression region? The authors can show more equivalent image for the lineage specification for better understanding of readers.”

Whether dCPECs arise from NECs asymmetrically or symmetrically is an interesting question, but one that our data cannot resolve, including the data on molecular coexpression or lack thereof. As mentioned for Q1 above, text about this limitation has been added (**line 402**).

Minor points:

“Q4. Line147: FigS2G is a typo of FigS2H?”

This typo has been corrected (**line 147**).

“Q5. Line230-: The authors discussed that the higher rate of induction of lateral ventricle type of ChP is consistent with “default” model, but I think this can be derived from the effect of BMP treatment because 4Vtype of ChP development process need Shh signal but the LV type need BMP signals. The authors can make discussion with this aspect.”

This is an astute point given the lack of any SHH-related reagents in the protocol. We have added this point (**line 231**) as well as one reference (**line 1305; reference 99**).

“Q6. In general, it is thought that the inhibition of SMAD signals induce NEC from pluripotent stem cells, but the authors added BMP4 from very early phase of NEC induction and some may think this is inconsistent with dual SMAD protocol. There are other protocols to induce neural fate by addition of BMPs from early phase of induction (Watanabe M et al. Stem cell reports 2022, Kuwahara A et al. Nature commn 2015), but is there some relationship between these articles with the author’s result regarding the mechanism to induce neural fate via BMP signals?”

Our protocol and its optimizations provide some insight, albeit limited, on these interesting questions about BMP signaling and neural induction, which have been added briefly in Discussion (**line 357**).

Additional supplemental data

Figure S8C,D: Expanded *in vitro* and *in vivo* series for C2 markers (FOXN4 and MCIDAS) – These data expand upon the timepoints and stages originally provided and further strengthen conclusions about C2 contraction:

- **Lines 617 (Methods), 1030 (Fig S8), and 1037 (Fig S8 legend)**

Figure S9L,M: Expanded *in vitro* series for C1a (CLDN5) and C1b markers (SC5D) as well as an expanded *in vivo* series with independent cytoplasmic markers for C1a (IGFBP7) and C1b (SERPINE2) – These data also expand upon the timepoints and stages originally provided and further strengthen conclusions about the timing of C1a and C1b differentiation:

- **Lines 1050 (Fig. S9) and 1075 (Fig. S9 legend)**